# A carbon-nitrogen negative feedback loop underlies the repeated evolution of cnidarian–Symbiodiniaceae symbioses

Guoxin Cui [1,5] ✉, Jianing Mi [2,3,5], Alessandro Moret[1,5], Jessica Menzies [1], Huawen Zhong[1], Angus Li [1], Shiou-Han Hung[1], Salim Al-Babili [2,4] & Manuel Aranda [1] ✉

Symbiotic associations with Symbiodiniaceae have evolved independently across a diverse range of cnidarian taxa including reef-building corals, sea anemones, and jellyfish, yet the molecular mechanisms underlying their regulation and repeated evolution are still elusive. Here, we show that despite their independent evolution, cnidarian hosts use the same carbon-nitrogen negative feedback loop to control symbiont proliferation. Symbiont-derived photosynthates are used to assimilate nitrogenous waste via glutamine synthetase–glutamate synthase-mediated amino acid biosynthesis in a carbon-dependent manner, which regulates the availability of nitrogen to the symbionts. Using nutrient supplementation experiments, we show that the provision of additional carbohydrates significantly reduces symbiont density while ammonium promotes symbiont proliferation. High-resolution metabolic analysis confirmed that all hosts co-incorporated glucose-derived $^{13}C$ and ammonium-derived $^{15}N$ via glutamine synthetase–glutamate synthase-mediated amino acid biosynthesis. Our results reveal a general carbon-nitrogen negative feedback loop underlying these symbioses and provide a parsimonious explanation for their repeated evolution.

The mutualistic symbiotic relationship between marine invertebrates and dinoflagellates in the family Symbiodiniaceae is one of the most common eukaryote-eukaryote endosymbiosis in our oceans and fundamental to coral reef ecosystems[1]. The symbiotic association with Symbiodiniaceae provides the hosts with photosynthetically derived carbohydrates and allows them to thrive in the oligotrophic environments of tropical oceans. Symbiodiniaceae symbioses have evolved convergently across a broad range of marine phyla, including single-celled foraminifera, sponges, cnidarians, platyhelminths, and mollusks[2,3]. Among these phyla, cnidarians have arguably evolved the largest diversity in Symbiodiniaceae symbioses. Two out of the four cnidarian subphyla that diverged >700 Mya[4,5], Anthozoa and Scyphozoa, have species that evolved symbiotic relationships with Symbiodiniaceae independently[6]. Anthozoans, which include octocorals, anemones, and reef-building corals, among others, have evolved the highest diversity in Symbiodiniaceae relationships[7].

The repeated evolution of these symbioses across such a diverse range of phyla suggests that a common mechanism might exist that regulates the interactions between hosts and symbionts. These interactions need to allow for bidirectional nutrient exchange to establish

[1]King Abdullah University of Science and Technology (KAUST), Biological and Environmental Science and Engineering Division, Red Sea Research Center, Thuwal 23955-6900, Saudi Arabia. [2]King Abdullah University of Science and Technology (KAUST), Biological and Environmental Science and Engineering Division, the BioActives Lab, Center for Desert Agriculture, Thuwal 23955- 6900, Saudi Arabia. [3]State Key Laboratory of Traditional Chinese Medicine Syndrome, The Second Affiliated Hospital of Guangzhou University of Chinese Medicine, Guangzhou, Guangdong, China. [4]King Abdullah University of Science and Technology (KAUST), Biological and Environmental Science and Engineering Division, the Plant Science Program, Thuwal 23955- 6900, Saudi Arabia. [5]These authors contributed equally: Guoxin Cui, Jianing Mi, Alessandro Moret. ✉e-mail: guoxin.cui@kaust.edu.sa; manuel.aranda@kaust.edu.sa

an environment conducive to symbiont growth and function, but at the same time, provide a mechanism to regulate and control symbiont proliferation within host tissues. Several possible mechanisms have been proposed and investigated in the past, including specific protein machinery that allows the host to directly interfere with the symbiont cell cycle[8,9], host-controlled preferential expulsion of dividing symbionts[10], as well as more simple nutrient-flux-based models[11,12]. However, the repeated evolution of these relationships across such a vast range of taxonomic groups makes simple models requiring fewer evolutionary novelties more likely. Previous studies in corals and anemones have suggested that symbionts are nitrogen-limited *in hospite*[13–15] and that their proliferation might be controlled via host-dependent ammonium assimilation[11,16]. A recent meta-analysis of transcriptomic data comparing symbiotic and aposymbiotic *Exaiptasia diaphana* revealed that symbiosis activates glutamine synthetase/glutamate synthase (GS–GOGAT) mediated amino acid biosynthesis in the host[11]. Metabolomic analyses further confirmed that symbiotic

anemones increase waste ammonium assimilation, likely due to the availability of symbiont-derived photosynthates[11,17]. Further, studies in corals have also found that the GS–GOGAT cycle is active in various coral species[18–21], and a recent single-cell RNA-seq study in the sea anemone *Exaiptasia diaphana* found that this pathway is activated in most cell types in response to symbiosis[22]. Moreover, it has been observed that the provision of organic carbon reduces symbiont densities in corals[23], providing additional support for the role of organic carbon availability in the regulation of symbiont density in zooxanthellate cnidarians. In conclusion, a simple chemostat model in which cnidarian hosts use the photosynthates provided by the symbionts to assimilate their own waste nitrogen via the GS–GOGAT cycle and its subsequent incorporation into non-essential amino acids appears to be the most parsimonious [Fig. 1, adapted from Cui, et al.[11]]. The model is based on a metabolic interaction that allows the host to convert sugar and waste nitrogen into valuable amino acids while simultaneously providing a mechanism for symbiont control without

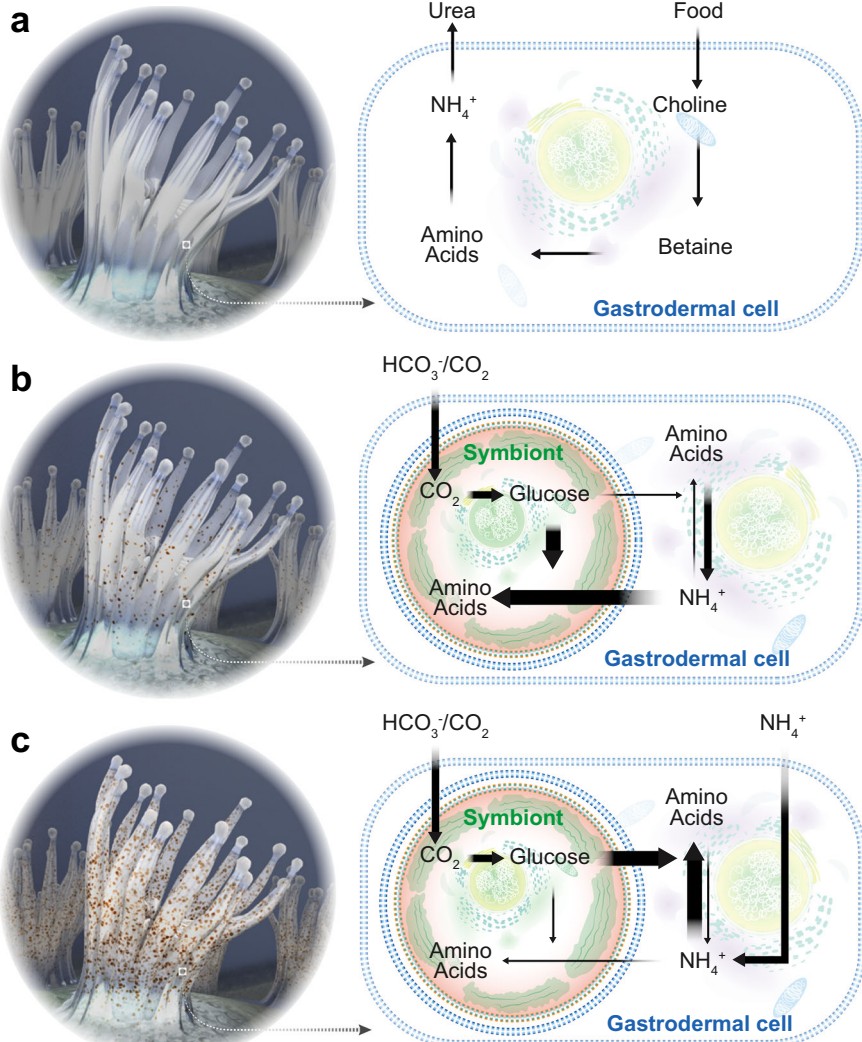

**Fig. 1 | Nutrient-flux-based negative feedback mechanism underlying symbiont population control. a** Aposymbiotic hosts are limited by the availability of energy-rich carbohydrates. They take up organic carbon from food and release nitrogenous waste to the surrounding environment as they do not have excess carbon backbones to assimilate the nitrogenous waste. **b** During the initial stages of symbiosis, a few colonizing symbionts have access to all host nitrogenous waste. This results in high nitrogen availability per symbiont that promotes symbiont cell proliferation. However, with increasing symbiont density, the competition for nitrogen increases until nitrogen becomes limiting and photosynthates are

produced in excess and translocated to the host. In response, the host experiences an increasing provision of energy-rich photosynthates from the symbionts while symbiont proliferation slows down gradually. **c** In fully symbiotic hosts, symbiont-provided glucose now provides an excess of carbon backbones that allow the host to assimilate a substantial amount of its nitrogenous waste. This further reduces nitrogen availability to the symbionts and decreases symbiont proliferation rates to eventually reach a balance between symbiont proliferation and symbiont decay. Adapted from Cui, et al. [11].

the requirement of additional means to regulate symbiont cell numbers.

While the model provides a parsimonious explanation for the regulation of these symbioses and their repeated evolution, an experimental demonstration and systematic evaluation across highly divergent cnidarian species is still missing. Here, we tested the model in three distantly related cnidarian species that evolved symbiosis independently, the reef-building coral *Stylophora pistillata*, the sea anemone *Exaiptasia diaphana*, and the upside-down jellyfish *Cassiopea andromeda*. Using $^{13}C$ and $^{15}N$ nutrient supplementation experiments combined with transcriptome profiling, enzyme activity assays, and high-resolution isotope tracing, we reveal the interplay between nutrient availability, symbiont density changes, and the co-incorporation of $^{13}C$ and $^{15}N$ via host-driven GS–GOGAT-mediated amino acid biosynthesis.

## Results

### Glucose and ammonium modulate symbiont density

Based on the model (Fig. 1), we hypothesized that symbiont cell density in symbiotic hosts is tightly controlled through a negative feedback response driven by the availability of glucose and ammonium. In this self-regulating system, increasing glucose level is expected to promote the capacity of the host to assimilate ammonium and, thus, limit nitrogen availability to the symbionts. This reduction in nitrogen availability would thus result in a decrease in symbiont cell density over time since the reduced nitrogen level would not be sufficient to support the original number of symbionts. Conversely, symbiont cell density is expected to increase when the availability of ammonium in the system increases.

To test this hypothesis, we manipulated the levels of glucose and ammonium in the surrounding environment of three cnidarian species and analyzed their symbiont density changes. As predicted, the supplementation with glucose resulted in significantly lower symbiont cell densities in the reef-building coral *Stylophora pistillata* (Fig. 2a), the sea anemone *Exaiptasia diaphana* (Fig. 2b), and the upside-down jellyfish *Cassiopea andromeda* (Fig. 2c). Conversely, symbiont cell densities increased significantly in *S. pistillata* and *E. diaphana* when ammonium was supplied, while *C. andromeda* showed an increasing but non-significant trend (Fig. 2). To exclude the possibility of an unspecific stress response as a cause for the decrease in symbiont densities, we performed additional experiments combining both glucose and ammonium. In line with our hypothesis, supplying both glucose and ammonium reversed the glucose-induced decrease in symbiont density in all three species (Fig. 2d–f). To verify these responses in an additional coral species, we repeated the experiments in the coral *Acropora hemprichii*, which showed the same responses as *S. pistillata* (Fig. 2g).

These responses confirmed that symbiont density is regulated by the availability of glucose and ammonium. However, the observation that the combined provision of glucose and ammonium did not restore symbiont densities to control levels in the *E. diaphana* and *C. andromeda* did not fully exclude the possibility that the observed reduction in symbiont densities in response to glucose could result from an unspecific stress response or a potential signaling role of glucose. To exclude this possibility, we repeated the combined glucose and ammonium supplementation tests in *E. diaphana* and *C. andromeda* using different concentrations of ammonium. As observed in our previous experiments, symbiont density negatively correlated with glucose concentration (Fig. 2h) and positively correlated with ammonium (Fig. 2i). Based on the results from dose-response experiments, we further increased the ammonium concentration and eventually restored the glucose-induced symbiont density decrease in *E. diaphana* (Supplementary Fig. 1a) and *C. andromeda* (Supplementary Fig. 1b) to control levels. These results confirmed that the observed symbiont density-reducing effect of glucose was not mediated by an unspecific stress response or a potential signaling role of glucose but

rather by a dose-response interaction as proposed by the metabolic model. Furthermore, the results suggested that both *E. diaphana* and *C. andromeda* have a substantially higher capacity for ammonium assimilation compared to *S. pistillata* as they required double the amount of ammonium to reach control symbiont densities.

### Glucose induces host-dependent ammonium assimilation

The concordant changes in symbiont density and nutrient availability provided strong support for the roles of glucose and ammonium in regulating symbiont density as predicted by the model. To gain further insight into the molecular pathway underlying this regulatory mechanism, we profiled glucose-induced transcriptomic changes in symbiotic and aposymbiotic *E. diaphana* as this species, in contrast to *S. pistillata* and *C. andromeda*, allows the direct comparison of symbiotic and aposymbiotic individuals.

Gene ontology (GO) assisted functional analysis of gene expression changes showed that glucose-induced genes were enriched primarily in amino acid biosynthesis, transcription regulation, and translation processes in both symbiotic and aposymbiotic anemones (Fig. 3a, b). The canonical Wnt signaling pathway, well-known for its function in cell proliferation[24], was activated in symbiotic *E. diaphana* supplied with glucose. This might indicate an overall upregulation of biological processes and pathways involved in cell growth upon glucose provision. Subsequent pathway enrichment analyses showed that glucose supplementation specifically induced the expression of genes associated with nitrogen metabolism and amino acid biosynthesis in both symbiotic and aposymbiotic *E. diaphana* (Fig. 3c). Further integration of these enriched biological processes and pathways highlighted the GS–GOGAT-mediated amino acid biosynthesis pathway that we identified previously[11] and all of the genes associated with this pathway were upregulated in symbiotic anemones compared to aposymbiotic ones when no glucose was supplied (Fig. 3d). It is interesting to note that glucose supplementation increased the expression of these genes in aposymbiotic anemones to levels comparable to those in symbiotic animals. Notably, we found that adding glucose to symbiotic *E. diaphana* upregulated the expression of these genes further.

We then tested if the host glutamine synthetase and glutamate synthase (GS–GOGAT) enzymes were actually active in the three species and if their activity changed in response to glucose and ammonium supplementation. We found that both host GS and GOGAT showed activity in all three species and treatments and that glucose generally promoted the activity of both enzymes while ammonium reduced it (Supplementary Fig. 2a–f). The overall pattern of enzyme activity changes aligned well with the pattern observed in glucose/ammonium-induced symbiont density changes: glucose generally enhances GS–GOGAT activity while decreasing symbiont density, whereas ammonium tends to suppress enzyme activities and elevate symbiont density.

### Host-dependent ammonium assimilation and amino acid synthesis

Based on the observed changes in gene expression and enzyme activity of GS and GOGAT, we hypothesized that host-driven ammonium assimilation via amino acid biosynthesis is likely the molecular pivot underlying symbiont population control, as previously proposed for *E. diaphana*[11]. To evaluate this hypothesis, we performed stable isotope tracer analysis using $^{13}C$ labeled glucose and $^{15}N$ labeled ammonium and ultra-high-performance liquid chromatography-high-resolution mass spectrometry (UHPLC-HR-MS).

Specifically, we examined the isotopic profiles of metabolic intermediates of the GS–GOGAT-associated amino acid biosynthesis pathways from host animals supplemented with $^{13}C_6$-glucose, $^{15}N$-ammonium, and $^{13}C_6$-glucose plus $^{15}N$-ammonium. The high-resolution mass spectra acquired with an enhanced resolution of 280,000 m/$\Delta$m

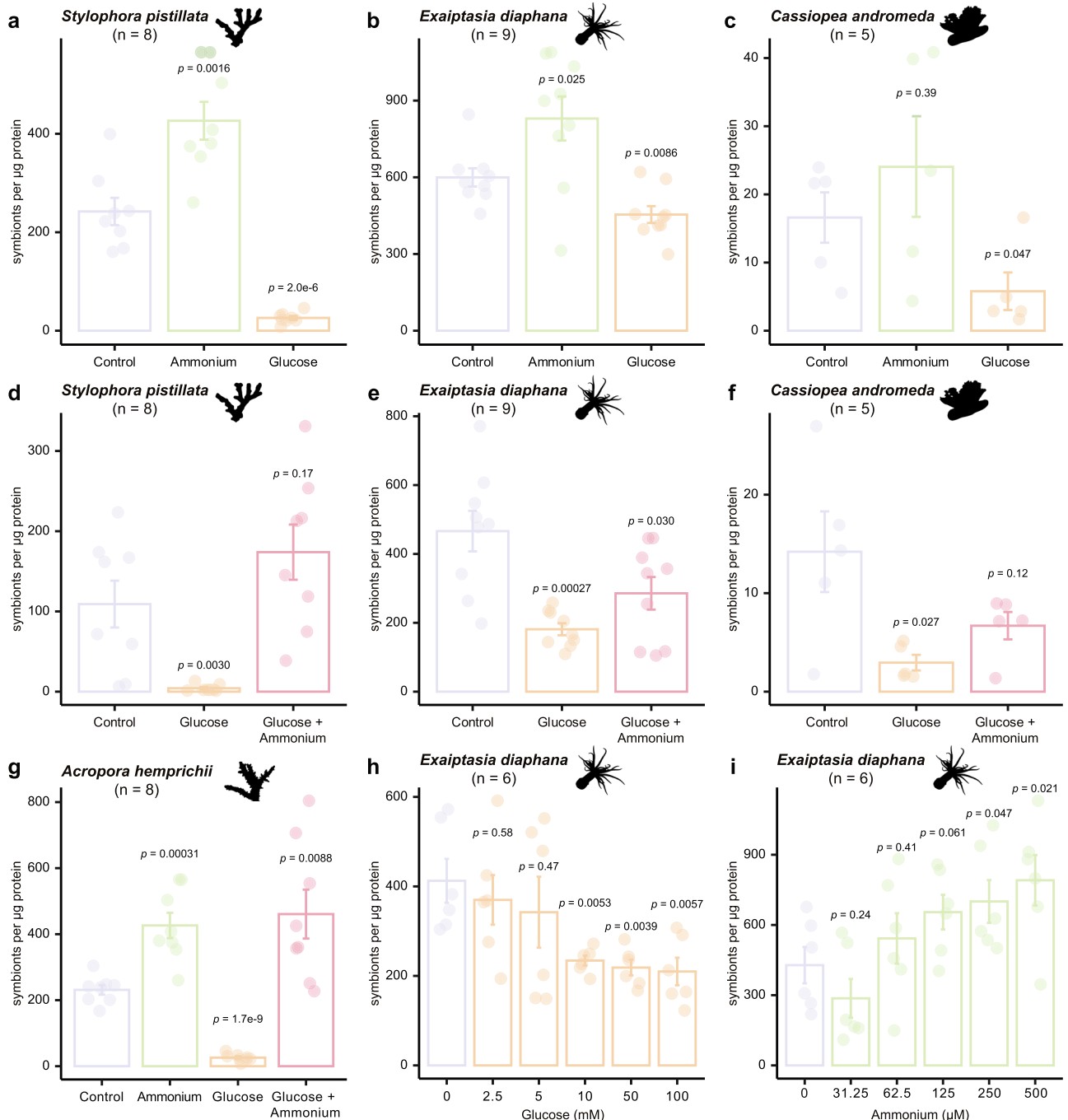

**Fig. 2 | Symbiont cell density changes induced by the availability of glucose and ammonium. a–g** The effects of 10 mM glucose, 250 μM ammonium, or both on symbiont density, represented by symbionts per μg host protein, were assessed in four cnidarian species: the coral *S. pistillata* (**a**, **d**), the sea anemone *E. diaphana* (**b**, **e**), the jellyfish *C. andromeda* (**c**, **f**), and the coral *A. hemprichii* (**g**). **h** Symbiont density changes in *E. diaphana* induced in response to different concentrations of glucose. **i** Symbiont density changes in *E. diaphana* induced in response to different concentrations of ammonium. For all plots, error bars represent the standard error of the mean. *p*-values were calculated using two-sided Welch's *t*-tests, comparing each condition to its respective control within each experiment. The *n* value indicates the number of biologically independent animals used in each experiment.

(at *m/z* = 200) facilitated the unambiguous identification and distinction of targeted compounds with different stable $^{13}C$ and $^{15}N$ isotopic compositions. Here we present the identification of glutamine from *E. diaphana* as an example (Fig. 4a). The natural isotopic distribution of glutamine standard shows two clear isotopic ions increasing by 1 amu, which are recognized as $[^{13}CC_4H_{10}N_2O_3]^+$ ion at *m/z* 148.07948 and $[C_5H_{10}^{15}NNO_3]^+$ ion at *m/z* 148.07321, respectively. In addition, their intensities are about 5% and 1% of the monoisotopic ion $[C_5H_{10}N_2O_3]^+$, respectively. Compared to the glutamine standard, the mass spectrum

of endogenous glutamine of *E. diaphana* incubated with both $^{13}C_6$-glucose and $^{15}N$-ammonium indicates the presence of various glutamine molecule compositions containing different amounts of $^{13}C$ and $^{15}N$ atoms (Fig. 4a). Following this strategy, we profiled the metabolites 3-phosphohydroxypyruvate, glutamate, glutamine, O-phospho-*L*-serine, serine, and glycine, which are intermediate metabolites of GS–GOGAT-mediated ammonium assimilation and subsequent amino acid biosynthesis pathways in *S. pistillata*, *E. diaphana*, and *C. andromeda* (Fig. 4b, Supplementary Figs. 3–25).

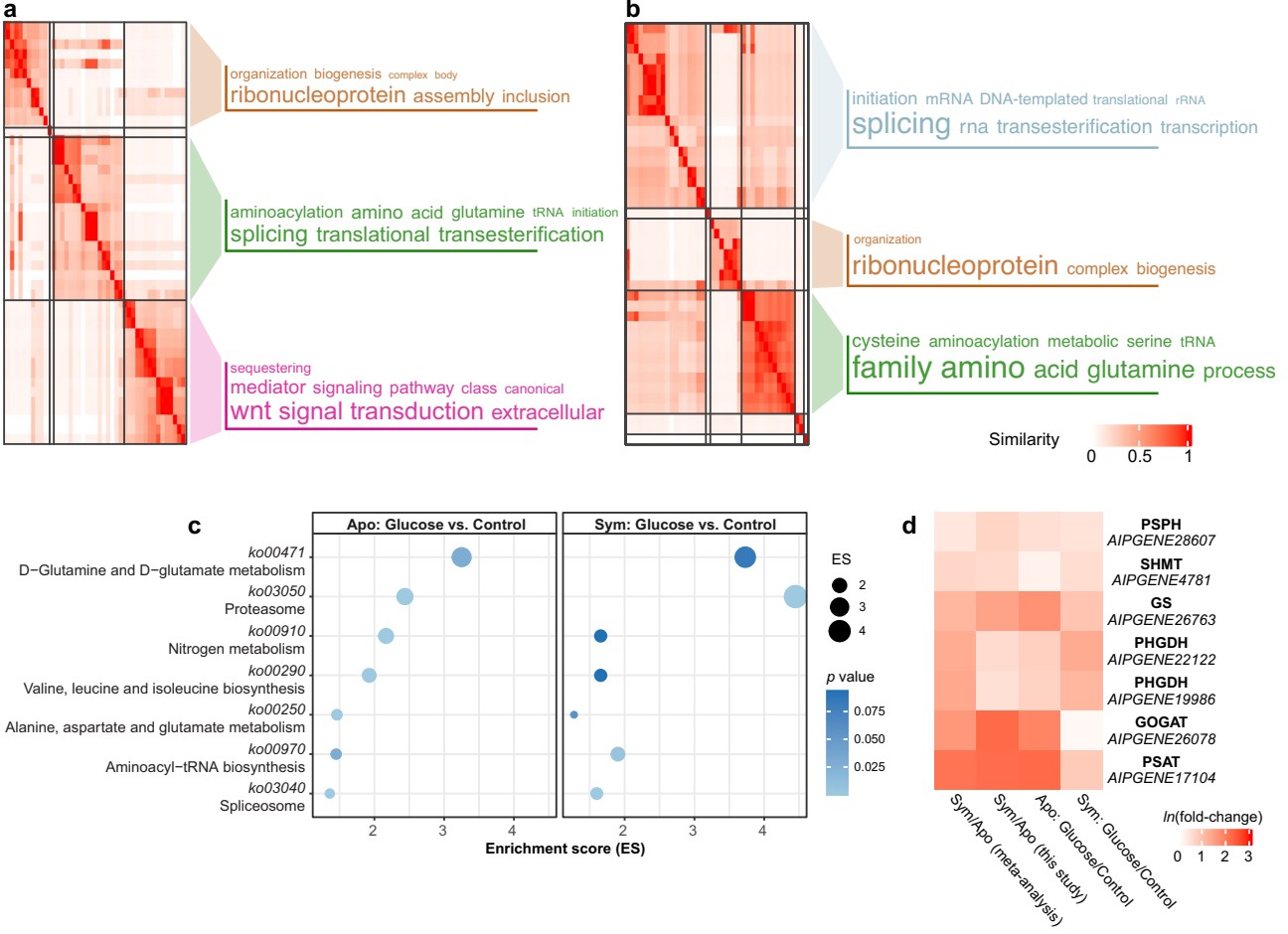

**Fig. 3 | Amino acid biosynthesis in response to glucose supplementation.**
**a**, **b** Biological process GO terms enriched for DEGs identified from the comparisons between symbiotic anemones incubated with or without 10 mM glucose (**a**) and aposymbiotic anemones incubated with or without 10 mM glucose (**b**). GO terms were clustered based on their semantic similarity. Word cloud summarizes the features with keywords in each GO cluster, with different font sizes representing the level of the enrichment. **c** Overrepresented pathways enriched in glucose-regulated DEGs in aposymbiotic and symbiotic *E. diaphana*. The enrichment score (ES) was calculated by dividing the actual by the expected number of DEGs associated with the corresponding pathway. The *p*-values were calculated with overrepresentation analysis implemented in the R package clusterProfiler. **d** Expression changes of hub genes involved in amino acid biosynthesis in response to symbiosis and glucose supplementation.

To examine the isotope integration in the different supplementation experiments, we first normalized the levels of isotopic ions to the abundance of the natural monoisotopic ion and then summarized them according to their isotopic compositions ($^{12}C^{14}N$, $^{13}C^{14}N$, $^{12}C^{15}N$, and $^{13}C^{15}N$).

In animals supplemented with $^{13}C_6$-glucose, the proportion of $^{13}C$-containing 3-phosphohydroxypyruvate, one of the intermediate metabolites derived from glycolysis, increased dramatically from the natural level (<5%) to >50% (mean percentage ± SE, 86.89% ± 0.88% in *S. pistillata*, 53.43% ± 6.97% in *E. diaphana*, and 69.75% ± 2.46% in *C. andromeda*). Similar isotopic results were observed in downstream metabolic intermediates of the amino acid biosynthesis pathway via the GS–GOGAT cycle (Fig. 4b).

The provision of $^{15}N$-ammonium in combination with $^{13}C_6$-glucose further increased the carbon isotope incorporation rate across all the intermediates in *E. diaphana* (84.53% ± 1.15%, $p = 0.0046$) and *C. andromeda* (91.20% ±1.29%, $p = 0.00039$), while there was no further increase observed for *S. pistillata* (84.43% ± 2.97%, $p = 0.45$). In addition, significant increases in $^{15}N$ incorporation rates were also observed when additional carbon was present. In the case where carbon is sourced from symbiont photosynthesis, symbiotic *E. diaphana* assimilated more $^{15}N$ compared to aposymbiotic anemones, highlighting the role of symbiont-derived photosynthates in host ammonium assimilation. In scenarios of glucose supplementation, $^{15}N$ incorporation rates were further enhanced. In particular, most of the $^{15}N$ isotope was identified in both $^{13}C$- and $^{15}N$-containing metabolites ($^{13}C^{15}N$) from animals with the combined treatment, while only a small proportion of the $^{15}N$ isotope ended up in $^{12}C^{15}N$ compounds (Supplementary Data 1–4), which indicates that most of the $^{15}N$ was assimilated through the incorporation into carbon backbones derived from the $^{13}C_6$-glucose provided. This finding further supported the hypothesis that the metabolization of glucose to 3-phosphohydroxypyruvate produces the carbon backbones required for ammonium assimilation through the GS–GOGAT cycle.

To further determine if the observed assimilation of ammonium is driven by the host animals, as suggested by the gene expression changes and enzyme activity assays, we examined the incorporation of $^{13}C$ and $^{15}N$ isotopes in aposymbiotic *E. diaphana* following the same experimental design (Fig. 4b). In line with the patterns observed in symbiotic animals, we found that 92.65% ± 0.79% of the isolated 3-phosphohydroxypyruvate contained $^{13}C$ isotope in aposymbiotic sea anemones. This proved that the $^{13}C$ isotope was integrated directly through the uptake and consumption of $^{13}C_6$-glucose by the host. Moreover, the downstream intermediate metabolites showed a considerable proportion of $^{15}N$-containing compounds ($^{12}C^{15}N$ and $^{13}C^{15}N$), especially in the $^{13}C^{15}N$ form (Fig. 4b,

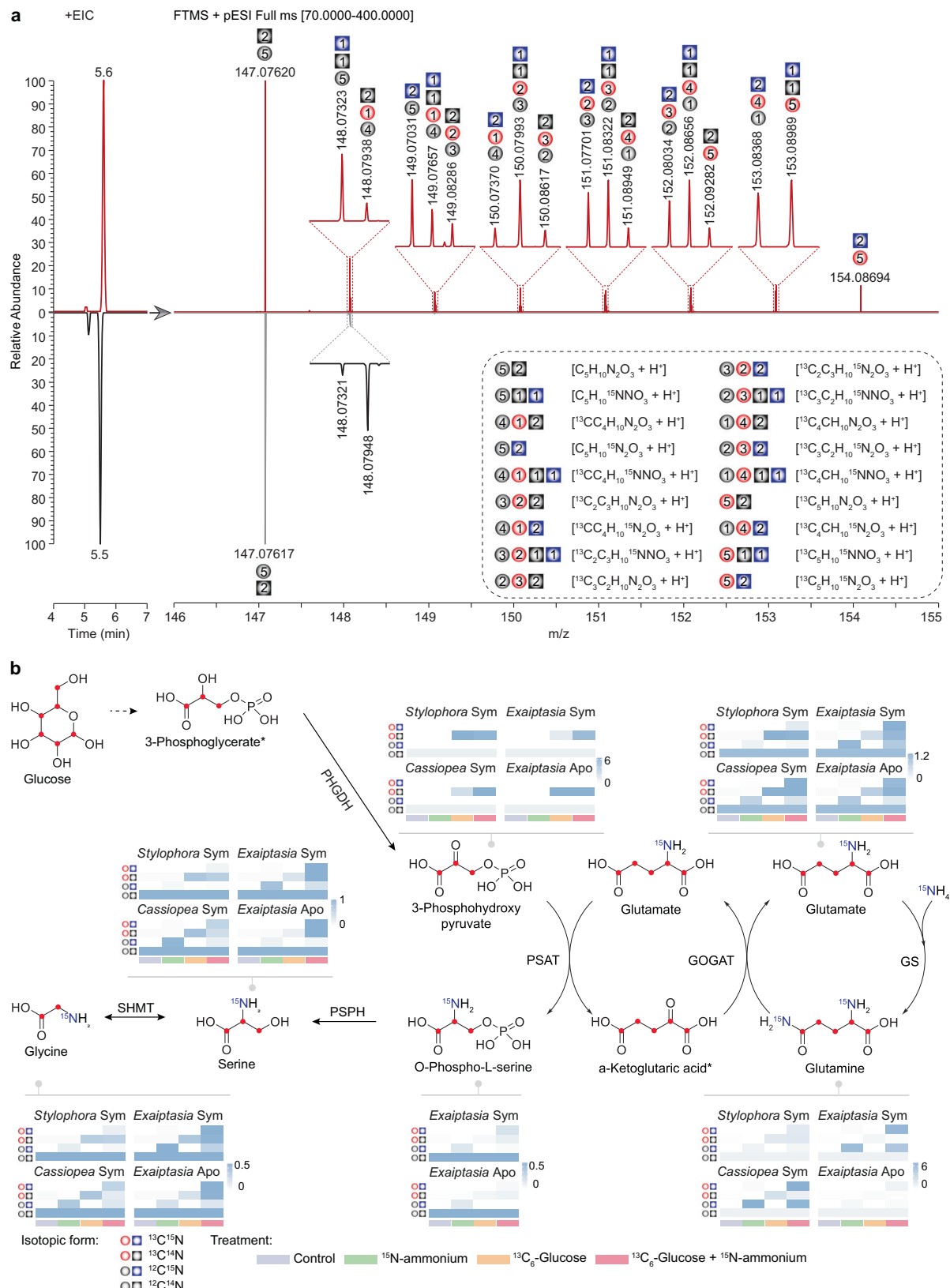

Supplementary Data 1–4). This provided further proof that the animal hosts assimilate the provided ammonium using carbon backbones derived from glucose metabolism, as hypothesized. The similar patterns observed between symbiotic and aposymbiotic sea anemones provide additional proof that the incorporation of ammonium into amino acids is driven by the host animals and that this process does not require the presence of symbionts as long as carbon backbones are provided. Although the presence of symbionts increased the co-incorporation of $^{13}C$ and $^{15}N$ in the combined treatment (Fig. 4b, Supplementary Data 1–4), the incorporation rates between symbiotic and aposymbiotic anemones were not significantly different ($t = -0.40$, $p = 0.69$).

**Fig. 4 | Identification of isotope-labeled metabolites using UHPLC-HR-MS. a** Extracted ion chromatograms (EIC, Left) and the isotopic distributions (Right) of glutamine from *E. diaphana* incubated with $^{13}C_6$-glucose and $^{15}N$-ammonium (Top) and the corresponding glutamine standard (Bottom). The inset corresponds to a zoom of the area in which different isotopologue compositions of glutamine (dashed box) were identified using HR-MS. The gray ball and square indicate $^{12}C$ atom and $^{14}N$ atom, respectively; the red ball indicates $^{13}C$ atom, the blue square indicates $^{15}N$, and the number of carbon and nitrogen atoms are inserted in the corresponding shapes. **b** Metabolic footprinting of stable isotopes in the three

selected cnidarian species. The proposed $^{13}C$ and $^{15}N$ isotope labeling is indicated as red dots or written in blue color in the structural formulas. Heatmap color indicates the relative abundance of isotope-labeled metabolites (specifically summarized as $^{13}C^{14}N$, $^{12}C^{15}N$, and $^{13}C^{15}N$) normalized to their non-labeled counterparts ($^{12}C^{14}N$). The isotopic forms of each compound shown in the heatmaps follow the same order as in the figure legend. The numeric values for heatmaps are included in Supplementary Data 5. Sym, symbiotic state; Apo, aposymbiotic state; *, undetectable metabolite.

## Metabolic destinations for glucose taken up by corals

Besides validating our hypothesis of host-driven ammonium assimilation as a universal mechanism in cnidarian symbioses, we also noticed that corals appear to differ in their use of glucose compared to sea anemones and jellyfish. To track the metabolic flux of $^{13}C$ in our target GS–GOGAT pathway, we calculated the proportional changes relative to total carbon atoms across the pathway metabolites (Fig. 5). *S. pistillata* started with significantly higher $^{13}C$ uptake (Welch's *t*-test: *S. pistillata* vs. *E. diaphana*, $p = 0.011$; *S. pistillata* vs. *C. andromeda*, $p = 0.00018$) but ended up with similar levels of $^{13}C$ in the endpoint metabolites compared to *E. diaphana* (Welch's *t*-test, $p = 0.87$) and *C. andromeda* (Welch's *t*-test, $p = 0.45$) when supplemented with $^{13}C_6$-glucose (Fig. 5a). This resulted in a significantly lower integration rate of $^{13}C$ into amino acids in *S. pistillata* (Supplementary Table 1), suggesting that corals use relatively more glucose for purposes other than amino acid biosynthesis. Furthermore, the simultaneous provision of $^{15}N$-ammonium and $^{13}C_6$-glucose did not increase the uptake of $^{13}C$ or its integration into amino acids in *S. pistillata* as observed for *E. diaphana* or *C. andromeda* (Figs. 4b and 5b). Conversely, *E. diaphana* and *C. andromeda* showed a higher relative capacity to integrate $^{13}C$ into amino acids, and this capacity was further enhanced when additional ammonium was provided.

## Discussion

Endosymbiotic relationships are the most intimate form of symbiosis, as the symbionts are maintained within the host's cells[25]. Naturally, this intimacy requires mechanisms that allow providing mutual advantages to both parties in order to maintain an evolutionary stable relationship while discouraging or penalizing parasitic traits that could destabilize the relationship and trigger a Red Queen's race between host and symbiont[26]. Oftentimes, this is prevented through the evolution of mechanisms that provide the host with means to control or limit symbiont proliferation in order to prevent the over-proliferation of symbionts at the host's expense[27,28].

Here, we tested our hypothesis that the widespread symbiotic relationships between cnidarians and their dinoflagellate symbionts in the family Symbiodiniaceae are based on a simple carbon-nitrogen negative feedback loop. This basic metabolic interaction allows cnidarian hosts to control symbiont proliferation in response to carbon availability while at the same time maximizing their capacity to assimilate and recycle scarce nitrogen waste into valuable amino acids. Briefly, carbon availability is tightly correlated with symbiont density since symbiont photosynthesis is the primary source of organic carbon for cnidarian hosts in the typical marine environment. Symbiont density generally correlates positively with photosynthetic production, with more symbionts providing more organic carbon to the hosts, and hence enhancing their ammonium assimilation capacity. This limits the nitrogen available for symbiont proliferation. In the case of a rapid increase in carbon availability, as in the glucose supplementation experiments shown here, a larger amount of ammonium in the system is assimilated by the host, which leaves symbionts in a more growth-limited state. This leads to a reduction of symbiont density over time as observed in this study and recently in the coral *S. pistillata*[23]. Conversely, when the negative feedback loop is imbalanced towards low carbon availability, host animals are not able to assimilate as much ammonium, and the

increasingly available nitrogen stimulates symbiont proliferation. In the natural environment, low carbon availability could result from low symbiont density, low photosynthetic efficiency in poor light conditions, or sudden increases in the ammonium concentration. In line with our model, these conditions have been shown to increase the symbiont population and hence re-balance the carbon-nitrogen equilibrium[14,29,30]. The simple model proposed here relies solely on a basic metabolic machinery that is generally present in cnidarians and, hence, does not require any species-specific evolution of novel mechanisms. It thus not only provides a simple metabolic mechanism for symbiont population control but also a parsimonious explanation for the repeated evolution of these symbiotic associations across many cnidarian taxa and potentially also other marine organisms.

While our results show that all four hosts employ ammonium assimilation to control symbiont proliferation, they also revealed important differences between the different cnidarian hosts and their use of glucose and ammonium. These differences have critical implications for the ability of the hosts to control their symbiont populations and, thus, stabilize the symbiosis. Specifically, we found that the rates at which these metabolites are taken up and metabolized differ substantially between the taxa studied, with the coral *S. pistillata* showing the highest uptake of $^{13}C_6$-glucose but the lowest relative incorporation rate of $^{13}C$ into $^{15}N$-containing metabolites. This difference in the incorporation of $^{13}C$ might result from physiological differences in carbon requirements and utilization, although other reasons, such as differences in symbiont type or photosynthate translocation from symbiont to host, cannot be excluded. However, reef-building corals, like *S. pistillata*, require a significant amount of glucose to meet the energy demands of the calcification processes. The significantly lower relative incorporation of $^{13}C$ in more downstream pathway metabolites in comparison to anemones and jellyfish might therefore suggest that corals use a larger part of the glucose, and the derived carbon backbones, to meet their energetic demands. Conversely, jellyfish and sea anemones do not calcify and do thus not have such a physiological demand. Hence, they are able to use more of the glucose provided for the assimilation of ammonium and subsequent amino acid biosynthesis. In the glucose treatment without additional ammonium, these non-calcifying species are rather nitrogen-limited as the provided glucose is sufficient to cover both the energetic demands as well as the assimilation of the waste ammonium available. However, the provision of both glucose and ammonium further promoted the uptake of both nutrients from the surroundings and subsequent ammonium assimilation and amino acid biosynthesis i.e., more glucose was taken up if additional ammonium was provided, which further increased ammonium assimilation and amino acid biosynthesis.

Our results, therefore, suggest that coral hosts might have a lower capacity to assimilate ammonium as they require more glucose to meet their energetic demands, which translates into a lower capacity to control nitrogen levels and, thus, symbiont proliferation. This lowered capacity could also result in a reduced ability to buffer imbalances in the availability of glucose and ammonium. As a consequence, the symbiotic relationship between corals and Symbiodiniaceae might be more sensitive to metabolic imbalances, i.e., changes in the fluxes of glucose and ammonium, which can result from reduced translocation

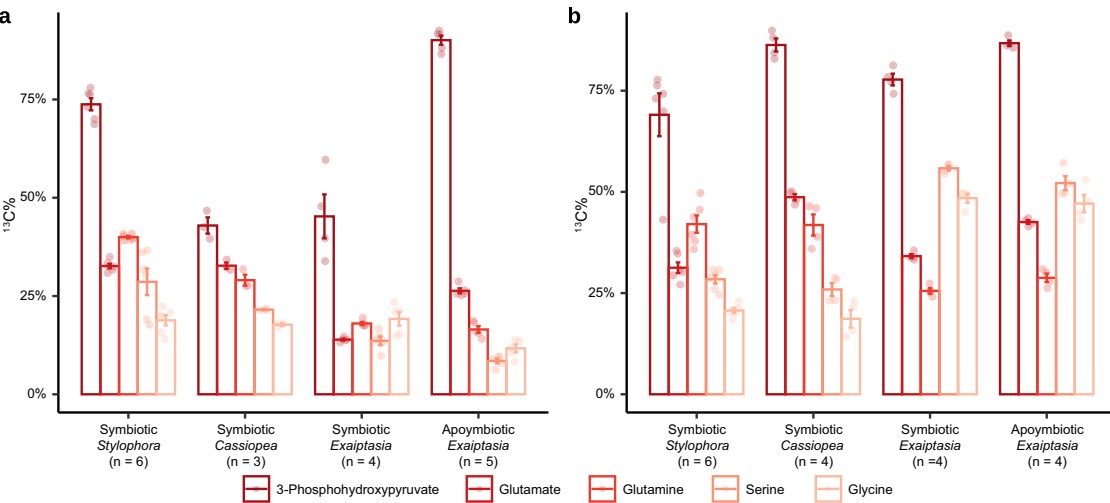

**Fig. 5 | The incorporation of $^{13}$C across metabolites of GS–GOGAT-mediated amino acid synthesis. a, b** $^{13}$C incorporation in response to supplementation with $^{13}$C$_6$-glcuose (**a**) or $^{13}$C$_6$-glucose + $^{15}$N-ammonium (**b**). The relative abundance of $^{13}$C was calculated for each intermediate metabolite associated with GS–GOGAT-mediated amino acid biosynthesis. Error bars represent the standard error of the average $^{13}$C percentage. *n* indicates the number of biologically independent animals used in each experiment. *p*-values were calculated using two-sided Welch's *t*-tests, comparing each condition to its respective control within each experiment.

of photosynthates during heat stress[21,31]. Interestingly, corals (*S. pistillata* and *A. hemprichii*) consistently showed the strongest and fastest response to glucose provision as reflected in a dramatic decrease in symbiont density. This quick response suggests that corals might possess a mechanism to actively reduce symbiont density by expelling symbionts when alternative carbon sources are available. Such a mechanism might allow corals to mitigate some of the drawbacks resulting from their reduced ability to control their symbiont population. In line with this, a recent study suggests that corals might also be able to control ammonium fluxes to the symbionts by varying the expression level of an ammonium transporter at the symbiosome membrane[32].

An obvious complication of our study is that metabolic signals are highly entangled in these endosymbiotic relationships, especially regarding ammonium assimilation as both partners possess the essential machinery. Ammonium assimilation by the symbionts and translocation to the host in the form of amino acids is currently assumed to be the main mechanism of nitrogen recycling for host animals. However, this assumption is challenged by increasing evidence that symbionts are nitrogen-limited in vivo[14,15]. A reasonable implication is that symbionts would fail to provide nitrogenous products in nitrogen-limited environments. We believe that the assumption that the symbionts are the main ammonium assimilators in cnidarian holobionts is based on their much higher capacity for ammonium assimilation per cell compared with host animals[19]. However, we showed in a recent study in *E. diaphana* that when accounting for the total biomass, both partners exhibit a comparable total capacity for nitrogen assimilation[22].

Our findings have important implications for our understanding of these symbioses as they shift our view of these relationships from being based on cooperation to being driven by the competition for nitrogen. This competition results in holobionts that are highly efficient with regard to nitrogen conservation and that function well within an equilibrium range defined by the capacity of the host and the symbiont to assimilate ammonium. This equilibrium range might also define viable host-symbiont combinations and, hence, contribute to the host-symbiont specificity.

Unlike many other nutritional endosymbioses that are driven by the complementation of the host's limited metabolic capabilities[33–35], our findings suggest that cnidarian hosts rely rather heavily on the provision of glucose to control their symbiont population. This host-

driven mechanism provides an effective metabolic strategy to gain control over the symbiotic relationship at the expense of being dependent on symbiont-derived glucose. This paradoxical interaction in which control over symbiont proliferation requires symbiont-derived photosynthates might also explain the general sensitivity of these relationships to environmental stressors that affect symbiont productivity or nutrient balance[36].

## Methods

### Cnidarian animals

Multiple colonies of the coral *Stylophora pistillata* and *Acropora hemprichii* were collected in the central Red Sea (Al Fahal Reef, 22°14′ 54″ N, 38°57′46″ E). The coral colonies were acclimatized in indoor tanks with constant sediment-filtered Red Seawater in-flow (salinity ~39-40 ppt) for at least three months before being used in this study.

The sea anemone *Exaiptasia diaphana* (strain CC7) was used in this study. Aposymbiotic *E. diaphana*, sea anemones free of symbionts, were generated following a cold-shock protocol[11]. Briefly, animals were treated at 4 °C for 4 h, followed by ~30 days of treatment in 50 μM diuron (Sigma-Aldrich) with daily water changes. At the end of the treatment, all anemones were individually analyzed via fluorescence microscopy to further confirm the absence of algal fluorescence. All animals, symbiotic and aposymbiotic, were then kept at 25 °C on a 12 h:12 h light:dark cycle with ~40 μmol photons m$^{-2}$s$^{-1}$ of photosynthetically active radiation and fed with freshly hatched brine shrimp, *Artemia*, approximately three times per week. The individuals used in this study were kept in such conditions for at least 6 months. The aposymbiotic anemones were individually inspected again right before experiments by fluorescence microscopy to confirm the free of algal symbionts.

The adult jellyfish *Cassiopea andromeda* were collected from the Red Sea (22°20′23.0″ N, 39°05′31.1″ E). The breeding pairs then spawned in the laboratory and fertilized in the autoclaved seawater. The embryos were transferred to a lab incubator at 26 °C and raised until the medusa stage. Different stages of *C. andromeda*, including polyp, ephyra, and medusa, were raised separately. All animals were maintained in autoclaved seawater at 26 °C on a 12 h:12 h light:dark cycle with ~40 μmol photons m$^{-2}$s$^{-1}$ of photosynthetically active radiation and fed daily with freshly hatched brine shrimp, *Artemia.* The jellyfish used in this study were kept in such conditions for at least six months.

Unless otherwise specified, the same light intensity and temperature mentioned above were used for all experiments. To avoid potential interference introduced by food, the animals were not fed over the course of the nutrient supplementation experiments.

## Glucose and ammonium supplementation

To examine how carbon-nitrogen balance regulates symbiont density, we incubated four cnidarian species at four nutrient conditions, including ambient seawater (control), glucose (10 mM), NH$_4$Cl (250 μM), and glucose (10 mM) plus NH$_4$Cl (250 μM). Initially, *S. pistillata*, *E. diaphana*, and *C. andromeda* were tested only in the first three conditions (Fig. 2a–c). To rule out stress as a cause of symbiont reduction in the glucose-treated animals, we performed an independent experiment with ambient seawater (control), glucose (10 mM), and glucose (10 mM) plus NH$_4$Cl (250 μM) (Fig. 2d–f). Since these two experiments were performed on different batches, factors such as differences in feeding regimen, light intensity, and water quality, among others, may contribute to variations in symbiont density even within the same species. To validate the observed pattern in a different coral species, we performed an additional experiment with all four treatments in parallel on *A. hemprichii* (Fig. 2g).

We chose these concentrations based on the estimated glucose and ammonium concentrations that are close to the buffering capacity of the GS–GOGAT metabolic model. The glucose concentration was determined based on the symbiont cell density usually found in sea anemones[37], the typical photosynthetic efficiency of symbionts *in hospite*[38], and the widely accepted fact that symbionts translocate over 90% of their photosynthates to host cells[39]. Initial estimation proposed that a fully symbiotic sea anemone would receive mg level of glucose from its endosymbionts every light cycle. Considering that the efficiency of glucose uptake from the environment would be lower than its intracellular translocation, we proposed that external supplementation would require a higher concentration of glucose to mimic the carbon supply from *in hospite* symbionts. To better determine the appropriate glucose concentration, we then incubated symbiotic sea anemones in a series of glucose treatments for 12 days and found that 10 mM is the lowest concentration that induces replicable symbiont cell density changes (Fig. 2h). The ammonium concentration was adapted from previous studies[14] and tested independently where we found that 250 μM is the lowest ammonium concentration to induce replicable symbiont cell density changes (Fig. 2i).

## Symbiont cell density measurement

**Corals.** Cell density changes in response to glucose and ammonium supplementation were measured for two coral species, *S. pistillata* and *A. hemprichii*, respectively. For each species, eight branches from different coral colonies were cut for each treatment. The coral branches were tied to plastic stands and placed into three transparent Nalgene™ straight-sided wide-mouth polycarbonate jars (Thermo Fisher). Then, 250 mL seawater from indoor acclimation tanks was used to fill up each of the jars and the water was changed every 2 days with fresh treatments applied. To ensure efficient gas exchange in these small volumes, we added magnetic stirring bars to the jars before placing them onto a Cimarec™ i Telesystem Multipoint Stirrer (Thermo Fisher) with constant stirring at 300 rpm. The whole setup was placed in an incubator at 25 °C with ~80 μmol photons m$^{-2}$s$^{-1}$ radiation and a 12 h:12 h light:dark cycle. After 12 days of incubation, coral fragments were airbrushed with a lysis buffer (0.2 M Tris-HCl, pH = 7.5; 0.5% Triton-X; 2 M NaCl) to dissociate and lyse the animal tissues. The tissue lysates were sheared by repeated passage through a 25G needle to release the symbionts. Then, 500 μL of each homogenized sample was centrifuged at 8000 × *g* for 2 min at room temperature.

**Sea anemones and jellyfish.** Nine polyps of *E. diaphana* and five polyps of *C. andromeda* were used for each of the three treatments: the ambient seawater control, 10 mM glucose, and 250 μM ammonium chloride (or 10 mM glucose plus 250 μM ammonium chloride in an independent experiment to test their combined effects). The incubation was performed in 6-well plates. Then, 8 mL autoclaved seawater with appropriate treatments was used and refreshed every 2 days. After 12 days of incubation, animal polyps were homogenized with the abovementioned lysis buffer using a cordless motor mixer (Thermo Fisher). The tissue homogenates were sheared by repeated passage through a 25G needle to release the symbionts. Then, 500 μL of each homogenized sample was centrifuged at 8000 × *g* for 2 min at room temperature.

**Cell counting.** For all the cnidarian animals, symbiont cells in the pellets were counted using a BD LSRFortessa™ Cell Analyzer (BD Biosciences) based on their chlorophyll fluorescence and forward-scatter signals (Supplementary Fig. 26). No obvious change in symbiont cell size or fluorescent intensity has been observed from flow cytometry profiling. Host proteins in the supernatants were quantified using a Pierce Micro BCA™ Protein Assay Kit (Thermo Fisher) according to the manufacturer's recommendations. Cell density was determined by normalizing the total cell number to the total host protein content. The normality of cell density data was tested using the Shapiro-Wilk test followed by Levene's test of homogeneity of variance. Statistical differences among conditions were calculated using two-tailed Welch's *t*-tests.

**Image analysis.** To verify the symbiont density changes across the different conditions, images of *E. diaphana* and *S. pistillata* were taken at the end of the 12-day incubation using a Leica DMI 3000B microscope or a Canon EOS Rebel t5i camera. Color images of coral fragments were first converted to 16-bit in Fiji v2.13.1[40]. Using the Color Inspector 3D plugin, lightness values from the Lab color space were extracted from four different spots for each coral fragment and from E4 square on the Coral Health Chart (CoralWatch) in the same image. The relative lightness of each coral fragment was then calculated by subtracting the E4 lightness from the average lightness of the coral fragment (Supplementary Fig. 27a). Fluorescent images of *E. diaphana* were analyzed using CellProfiler v4.2.5[41]. Briefly, the images were first cropped to retain ~100 × 200 pixels from the base of the tentacle in focus for each sea anemone. Symbionts were identified based on their fluorescent signal and counted automatically. Both symbiont counts and pixel numbers for the cropped region were extracted. The symbiont density was then calculated by normalizing cell numbers to areas calculated from pixels (Supplementary Fig. 27b). Symbiont density changes aligned well with cell densities determined by flow cytometry-based cell counts and per protein content normalization. However, due to the low throughput and limited cross-species interpretation of imaging-based measurements, we adopted the FACS-based method for all cell density analyses.

## *E. diaphana* RNA-seq

To test the effect of glucose supplementation on gene expression, we compared the transcriptomic profiles of glucose-incubated (Gluc) symbiotic (Sym) and aposymbiotic (Apo) sea anemones with non-treated control (Con) animals. Experiments were performed in six-well plates. Sym and Apo animals were incubated with 8 mL of seawater supplied with or without 10 mM glucose. Seawater was changed every 2 days with the addition of a new glucose dose to appropriate wells after each water change. As previously described[22], five plates representing five independent biological replicates were processed simultaneously. The experiment ran for 11 days, with anemones being collected on the last day at 11:00 a.m., ~6 h into the light period. The collected animals were snap-frozen in liquid nitrogen and kept at −80 °C until further processing for total RNA extraction.

The total RNA was extracted using the RNeasy Mini Kit (QIAGEN) following the manufacturer's protocol for extraction from animal tissue. RNA quality was then assessed on an Agilent 2100 Bioanalyzer using the Agilent RNA 6000 Nano kit. RNA-seq libraries were prepared using the TruSeq RNA Library Preparation Kit v2 (Illumina) according to the manufacturer's protocol. All 20 libraries were then pooled and sequenced on an S1 flow cell with an Illumina Novaseq 6000.

RNA-seq reads were mapped against the revised *E. diaphana* gene models to quantify their expression levels using kallisto v0.44.0[42]. Differentially expressed genes (DEGs) were identified using sleuth v0.29.0[43] in two pairwise comparisons (SymGluc vs. SymCon and ApoGluc vs. ApoCon). Gene Ontology (GO) term enrichment analysis and Kyoto Encyclopedia of Genes and Genomes (KEGG) pathway enrichment analysis were conducted on the DEGs using topGO v2.52.0[44] or clusterProfiler v4.8.1[45], respectively. The enriched GO terms were further summarized and visualized using simplifyEnrichment v1.10.0 based on semantic similarity[46].

### Enzyme activity assay

To assess the effects of nutrient supplementation on the key enzymes, we measured the activities of GS and GOGAT in *S. pistillata*, *E. diaphana*, and *C. andromeda*. The same experimental setups as in the symbiont density measurement were followed here, except that only three biological replicates were used for each of the four conditions: control, 10 mM glucose, 250 µM ammonium, and 10 mM glucose plus 250 µM ammonium. After 12 days of treatment, the animals were collected and lysed as described above. The host protein content of each sample was then measured using a Pierce Micro BCA™ Protein Assay Kit (Thermo Fisher). GS activity was measured using a Glutamine Synthetase Activity Assay Kit (ab284572, Abcam). GOGAT activity was measured using a Glutamate Synthase Microplate Assay Kit (CAK1064, Cohesion Biosciences). One unit of enzyme activity was then defined as the amount of enzyme that produces 1 nmol of ADP (for GS) or decomposes 1 nmol of NADH (for GOGAT) per minute at pH 7.5 at 26 °C. The calculated enzyme activities were then normalized to the corresponding host protein contents.

### Isotope labeling and metabolite extraction

To track the uptake and incorporation of $^{13}C$ and $^{15}N$ isotopes, *S. pistillata* fragments, symbiotic and aposymbiotic *E. diaphana* polyps, and *C. andromeda* at the medusa stage were incubated for 48 h with either filtered seawater, filtered seawater with 10 mM $^{13}C_6$-glucose, filtered seawater with 250 µM $^{15}N$-ammonium, or filtered seawater and 10 mM $^{13}C_6$-glucose and 250 µM $^{15}N$-ammonium. After 48 h of incubation, the animal tissues were homogenized following the same procedure mentioned above. The homogenates were centrifuged at 10,000 × *g* for 5 min at 4 °C. Pellets containing the symbionts were discarded while supernatants containing animal tissue lysates were transferred to new tubes on ice. The same centrifugation process was repeated three times to ensure the complete removal of symbionts. Supernatants of several representative samples from the last centrifugation were inspected by fluorescence microscopy to confirm the absence of algal chlorophyll. Animal tissue lysates in the supernatants were snap-frozen using liquid nitrogen. Animal metabolites were then extracted from the symbiont-free animal tissue lysates as previously described[17]. Briefly, animal tissue homogenates were further lysed in 5 mL milli-Q water and lyophilized using a freeze dryer (Labconco). The lyophilisates were resuspended in 1 mL pre-chilled (−20 °C) 100% methanol, sonicated for 30 min at 4 °C in an ultrasonication bath (Branson), and centrifuged at 3000 × *g* for 30 min at 4 °C. The supernatants were collected and stored at −80 °C. The pellets were resuspended in 1 mL 50% methanol (−20 °C) and centrifuged at 3000 × *g* for 30 min at 4 °C. The supernatants were then combined with those collected from the previous step. The total extracts were then centrifuged at 16,000 × *g* for 15 min at 4 °C to remove any potential particulates. The

supernatants were dried using a speed vacuum concentrator (Labconco) and stored at −80 °C until further processing.

### Ultra-high-performance liquid chromatography-high-resolution mass spectrometry

Amino acid standard solutions were prepared by diluting Amino Acid Standard H (Thermo Fisher) to the concentrations of 25 µM for all amino acids except *L*-cystine (12.5 µM), followed by a 10-fold dilution with 25% aqueous methanol. Standard solutions for 3-phosphohydroxypyruvate (Sigma-Aldrich) and O-phospho-*L*-serine (Sigma-Aldrich) were individually prepared at a concentration of 2.5 µM in 25% aqueous methanol. The solutions for host metabolites were prepared with 200 µL of 25% aqueous methanol and filtered with a 0.2 µm filter before the UHPLC-HR-MS analysis.

Detections of the amino acids and intermediates (3-phospho-hydroxypyruvate, and O-phospho-*L*-serine) were performed on a Dionex Ultimate 3000 UHPLC system coupled with a Q Exactive Plus mass spectrometer (Thermo Fisher) with a heated-electrospray ionization source. Chromatographic separation of amino acids and intermediates was carried out on an ACQUITY UPLC® BEH Amide column (130 Å, 1.7 µm, 2.1 mm × 100 mm, Waters) maintained at 35 °C. The mobile phases A (water/formic acid, 100/0.1, v/v) and B (acetonitrile/formic acid, 100/0.1, v/v) were employed for eluting amino acids at a flow rate of 0.25 mL/min and with the gradient program: 0–8 min, 95% B to 25% B; 8–11 min, 25% B; 11–12 min, 25% B to 95% B; 12–15 min, 95% B. In addition, intermediates were eluted with the gradient program: 0–5 min, 100% B to 25% B; 5–8 min, 25% B; 8–9 min, 25% B to 100% B; 9–12 min, 100% B. The injection volume was 2 µL. Amino acids were detected using a mass spectrometer operated in positive mode with a spray voltage of 3.0 kV, sheath gas flow rate of 35 arbitrary units, auxiliary gas flow rate of 10 arbitrary units, spray capillary temperature of 300 °C, auxiliary gas heater temperature of 325 °C, AGC target of 3e6, and resolution of 280,000. In addition, intermediates were detected using a mass spectrometer operated in negative mode with a spray voltage of 2.5 kV, sheath gas flow rate of 40 arbitrary units, auxiliary gas flow rate of 20 arbitrary units, spray capillary temperature of 325 °C, and auxiliary gas heater temperature of 350 °C. In this work, Xcalibur software was used for MS data acquisition, peak identification, signal extraction, and related quantifications.

### Metabolite identification and quantification

Amino acids and intermediates from animal tissues were identified and assigned based on their accurate mass and matching with the corresponding standards. Isotopologues of each metabolite were then extracted and quantified by their peak areas. Isotopologue abundances quantified from control samples incubated with ambient seawater were used as the baseline to calculate the ratios of $^{13}C$- and/or $^{15}N$-containing isotopologues ($^{13}C^{14}N$, $^{12}C^{15}N$, and $^{13}C^{15}N$) relative to their natural non-labeled forms ($^{12}C^{14}N$). The normalized ratios were then used in the following comparative analyses.

To examine the efficiency of $^{13}C$ incorporation in different hosts, we calculated their $^{13}C$ percentages in total carbon element for each GS–GOGAT-associated metabolite. The relative incorporation rates were then calculated by comparing $^{13}C$ percentages between the start point and the endpoint of the pathway (Supplementary Table 1).

### Reporting summary

Further information on research design is available in the Nature Portfolio Reporting Summary linked to this article.

## Data availability

All quantitative results extracted from UHPLC-HR-MS analysis are provided as Supplementary Data files. The MS raw data generated in this study have been deposited in the NIH Common Fund's National

Metabolomics Data Repository[47] under accession code ST002870. RNA-seq data have been deposited in the NCBI Sequence Read Archive under accession codes PRJNA1018325 and PRJNA879277. Source data are provided with this paper.

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

## Acknowledgements

This work was supported by baseline funding from King Abdullah University of Science and Technology (KAUST) to M.A. and S.A.-B. Figure 1 was created by Heno Hwang, scientific illustrator at King Abdullah University of Science and Technology (KAUST).

## Author contributions

M.A. and G.C. conceived the study. A.M., G.C., and A.L. performed the nutrient supplementation experiments and the algal cell density measurements. J.Me. and G.C. performed the RNA-seq experiment and related data analysis. A.M. and G.C. performed the enzyme activity assay. G.C. and A.M. performed the isotope-labeling experiments and extracted the metabolites. J.Mi. and S.A.-B. performed UHPLC-HR-MS experiments. G.C., J.Mi., A.M., and H.Z. analyzed the metabolomic data. S.H.H. raised the jellyfish line. G.C. and M.A. wrote the initial draft of the manuscript with input from all authors. All authors reviewed and edited the manuscript.

## Competing interests

The authors declare no competing interests.
