## [Peer Review File · Nature Communications]

A carbon-nitrogen negative feedback loop underlies the repeated evolution of cnidarian-Symbiodiniaceae symbiosesREVIEWER COMMENTS

Reviewer #1 (Remarks to the Author):

This manuscript by Cui et al. investigated the impacts of supplying glucose, ammonium, or both to different cnidarian species. They analyzed the isotopic profiles of the amino acids from host animals supplemented with $^{13}\text{C}_6$ -glucose, ^{15}N -ammonium, or combined $^{13}\text{C}_6$ -glucose and ^{15}N -ammonium. This work was centered around the hypothesis that the symbiont density is regulated by the integration of fixed carbon from the symbionts and ammonium from the host, which has already been demonstrated in a few recent studies in sea anemones and corals. While it is important to understand the ecological and evolutionary aspects of this critical symbiosis for coral reef ecosystems, this manuscript appeared to expand to additional cnidarian species.

The authors tested the impacts of supplying glucose, ammonium, or both (Figure 2, and Figures S1&S2), but I have some concerns regarding the data interpretation, and think that it would be beneficial to consider alternative explanations for the data: (1) Glucose supplementation to the symbiotic animals caused the reduction of symbiont density, and the authors hypothesized that the reduction is due to a negative feedback response driven by the availability of glucose and ammonium. The authors talked about the model of the incorporation of C and N into host metabolites via GS/GOGAT cycle, and it would be more convincing if the activity of GS/GOGAT were measured for the experiments in Figure 2 and Figures S1&S2. (2) In addition to its nutritional role, glucose also acts as a signaling molecule involved in various processes. It would be beneficial to include an alternative hypothesis that glucose may trigger dysbiosis, thereby resulting in a reduced symbiont density. (3) The fact that supplementing ammonium besides glucose did not fully restore the loss of symbiont density in *Aiptasia* and *Cassiopeia* (Figure S1) further suggests that glucose may have additional effect(s) to the symbiotic relationships.

In addition, here are some more comments to further improve the manuscript.

Figure 2 showed the symbiont cell density changes induced by the availability of glucose and ammonium. "Greek letters indicate statistical differences with a significance cut-off at $p <$

0.05.” - which samples were conducted for the statistical test?

Figure S1 showed the combined effects of glucose and ammonium on symbiont cell density changes. Was there any particular reason to separate this data from Figure 2? Logically it should be included in Figure 2. One thing to note is that the symbiont density in the control samples was a little bit different for *Aiptasia* and coral in those two figures.

Figure 3: It would be beneficial to label the metabolites clearly in the figure... and in its current form this figure is confusing... While obviously ^{13}C -glucose and ^{15}N -ammonium were incorporated into the host metabolites, which is not surprising, it is unclear how it is linked to regulating symbiont density. The authors may want to clarify it in the results and discussion.

Line 190 and Figure 4: The authors mentioned “host-driven ammonium assimilation”, however, it should be pointed out that the symbiont algae can also assimilate nitrogen and provide amino acids to the host. In particular, figure 4 showed the differences in the incorporation of ^{13}C between symbiotic and symbiont-free animals, and including the role of the algal symbiont would enrich the discussion.

Line 262: “our findings suggest that cnidarian hosts rely rather heavily on the provision of glucose to control their symbiont population”, again, alternative hypotheses, for example, the signaling role of glucose, would be beneficial for interpreting and discussing the results.

Reviewer #2 (Remarks to the Author):

The manuscript uses well-controlled stable isotope labelling to explore the mechanism(s) of symbiont population control and organic C and inorganic N assimilation in the cnidarian-dinoflagellate symbiosis. This is important work both for advancing symbiotic theory but also in the contexts of holobiont stability in response to environmental perturbation (e.g. coral bleaching). The work is methodologically sound and well performed, and the manuscript well written. My concerns are with the novelty and interpretation of the dataset, and the paper may be acceptable after addressing the following questions:

Please state clearly the novelty of this dataset relative to a variety of papers investigating C/N uptake dating back to the 1980s. The hypothesis tested, nitrogen as limiting to symbiont population growth, and organic C supplementation resulting in large declines in symbiont population is not novel. This work tests it elegantly and with more breadth and resolution (the MS work is strong!), but the novelty must be very clearly stated to differentiate it from past work.

My understanding was that ammonium assimilation in the coral symbiosis was primarily via GS/GOGAT in the symbionts, not the host (see Pernice et al 2012 ISME, A single-cell view...). This is mentioned only in passing. How does algal ammonium uptake (and potential diversity in ammonium affinity amongst symbiont species) affect your model?

The supposition on different degrees of ^{13}C glucose incorporation in different taxa (e.g. line 227 forward) is weakened by only having 1-2 members of each taxon, limiting the ability to interpret the data. I noticed replication in another coral, *Acropora*, in the supplemental data (Fig S2). This should absolutely be moved into the main text and figures to strengthen the manuscript. However, it should still be noted that there are not enough representative taxa ($n = 1-2$) to be conclusive about entire subphyla. I would refer to the organisms by the genus name throughout the manuscript, rather than “anemones” or “jellyfish” as it implies that these patterns would necessarily hold throughout the groups.

Ln174: “To further determine if the observed assimilation of ammonium is driven by the host animals...” I believe here you’ve shown that the (aposymbiotic) host can assimilate ammonium, especially when supplemented with glucose, but that doesn’t mean that much ammonium is assimilated by the host when in symbiosis, as opposed to by the symbionts. Again, how does algal assimilation of ammonium complicate your analysis? Could the ^{15}N you’re detecting arise from algal assimilation and translocation of N-containing compounds to the host (amino acids etc.) and/or secondary assimilation of after catabolism of those compounds?

More minor concerns:

- The title of the paper is somewhat misleading, in that it implies that this is an evolutionary

study when it is not.

- Ln233: “This difference in the incorporation of ^{13}C might result from physiological differences in carbon requirements and utilization...” This seems vague, and I’m not sure how to interpret it.

- Ln310: What is meant by “a jointed experiment with four treatments in parallel”? Are these data meant for another manuscript?

- Fig. 3 is elaborate and I appreciate the amount of effort that went into it, but I find the heat maps to be quite unintuitive. Is there a more direct way to display these data?

Reviewer #3 (Remarks to the Author):

The manuscript by Cui et al. describes an elegant study that provides experimental evidence for the hypothesis that a negative carbon-nitrogen feedback loop is responsible for tightly controlled symbiont cell number in a variety of cnidarian host species. To this end, the authors tested the predictions of the model in three different cnidarian model systems that independently evolved symbiosis with Symbiodiniaceae. First, they administered exogenous glucose and ammonium to the animals and observed the numbers of the symbionts. These experimental data were supplemented by extensive stable isotope analyses. These data support the negative feedback model based on nutrient flux.

Specific comments:

Line 57: Please explain GS/GOGAT.

Figure 1: I think, it would be helpful to include symbiont proliferation to the model, as it is your read out in your experiments.

Figure 2. Please provide in the legend the concentrations of ammonium and glucose used.

Figure 2: To show these effects more convincingly, I propose to show that the effects of glucose and ammonium on symbiont cell numbers are concentration dependent as predicted; at least for one model species.

Figure 2: Why is the number of symbiont cells normalized to μg of protein rather than host cells? Based on your model, I would guess that protein content is not a good choice for normalization because AS synthesis is not independent in your experiment. It is directly dependent on the amount of glucose and ammonium provided. Please provide additional evidence of the changed symbiont cell numbers, e.g., by microscopic analyses of individual host cells.

Line 108; Figure S1: Please be precise here, in *E. diaphana* the effect was significant.

Line 293: Please describe the exact conditions for the treatment, timing, light conditions, feeding, etc.

Line 306: Please show the data in the supplements.

Discussion: Given the effects of glucose and ammonium on symbiont cell number, it would be relevant to discuss the mechanisms and timing of cell number adjustment. How quickly do the changes occur, especially the reduction in symbiont cells? Do symbiont cells die or are they just not proliferating? How long does a symbiont cell live? Do they get expelled from the host? I would encourage the author to discuss these points in more detail.

Response to Reviewer Comments

We would like to thank the reviewers for their constructive comments and suggestions. We have addressed all comments in detail below and we hope that our changes, clarifications, and additional experiments resolve all remaining concerns. Having said that, we would like to start by providing some general clarification on the critical issue of nitrogen recycling in the cnidarian holobiont.

Both host and symbionts have the enzymatic machinery and capacity to recycle waste ammonium into amino acids. This makes it nearly impossible to separate and quantify the contributions of host and symbionts regarding ammonium recycling. A substantial amount of the literature describes the symbionts as the main holobiont unit of ammonium assimilation, and it is assumed that the recycled nitrogen is translocated back to the host in form of amino acids. However, this has never been quantified appropriately due to the mentioned difficulty of separating the contributions of host and symbionts. It is well accepted, however, that symbionts are nitrogen limited in symbiosis and there is ample experimental evidence provided both in this study as well as the literature. It should, however, be noted, that the assumption that the symbionts are the main unit of nitrogen recycling, and that this nitrogen is returned to the host in form of amino acids is not compatible with the fact that the symbionts are nitrogen limited in symbiosis. They can, in fact, not be the main unit of nitrogen recycling and be nitrogen limited at the same time. In other words, if their role was to recycle the ammonium for the holobiont then supplementation with ammonium should lead to increased production and translocation of amino acids to the host. However, this is not what we and others have continuously observed.

Furthermore, our and other studies have shown that host ammonium assimilation is induced in symbiotic anemones and coral (Lehnert et al. 2014 G3, Cui et al. 2019 PLoS Genetics, Bednarz et al. 2020 Aquatic Toxicology). In fact, aposymbiotic anemones release nitrogen to the environment in form of ammonium and urea (Wilkerson and Muscatine 1984 Proc. R. Soc. Lond. B.). However, upon becoming symbiotic, they induce the expression of the GS/GOGAT pathway (Lehnert et al. 2014 G3, Cui et al. 2019 PLoS Genetics, but also see new Fig. 2 & 3), while, at the breakdown of symbiosis as triggered by heat stress, they suppress the expression of GS and GOGAT (Rädecker et al. 2021 PNAS). We would further like to stress the point that we developed our initial model (including the underlying molecular pathway) based on a meta-analysis of four independent transcriptome studies comparing symbiotic and aposymbiotic *E. diaphana* (Cui et al. 2019 PLoS Genetics). Here, we went on to test this model in other cnidarian species. This is also mentioned in the introduction in lines 53 – 62. We now also provide additional experiments showing that host GS and GOGAT are active in all three species and that the activity generally increases in response to glucose or glucose + ammonium supplementation (new Figure S2). This provides additional experimental evidence for the activation of this pathway in the host in symbiosis.

We believe that the combined evidence, together with the fact that this metabolic pathway presents the most parsimonious model for symbiont density control (no further protein machinery required), provides compelling evidence for our model.

Reviewer #1 (Remarks to the Author):

This manuscript by Cui et al. investigated the impacts of supplying glucose, ammonium, or both to different cnidarian species. They analyzed the isotopic profiles of the amino acids from host animals supplemented with ¹³C6-glucose, ¹⁵N-ammonium, or combined ¹³C6-glucose and ¹⁵N-ammonium. This work was centered around the hypothesis that the symbiont density is regulated by the integration of fixed carbon from the symbionts and ammonium from the host, which has already been demonstrated in a few recent studies in sea anemones and corals. While it is important to understand the ecological and evolutionary aspects of this critical symbiosis for coral reef ecosystems, this manuscript appeared to expand to additional cnidarian species.

The authors tested the impacts of supplying glucose, ammonium, or both (Figure 2, and Figures S1&S2), but I have some concerns regarding the data interpretation, and think that it would be beneficial to consider alternative explanations for the data: (1) Glucose supplementation to the symbiotic animals caused the reduction of symbiont density, and the authors hypothesized that the reduction is due to a negative feedback response driven by the availability of glucose and ammonium. The authors talked about the model of the incorporation of C and N into host metabolites via GS/GOGAT cycle, and it would be more convincing if the activity of GS/GOGAT were measured for the experiments in Figure 2 and Figures S1&S2. (2) In addition to its nutritional role, glucose also acts as a signaling molecule involved in various processes. It would be beneficial to include an alternative hypothesis that glucose may trigger dysbiosis, thereby resulting in a reduced symbiont density. (3) The fact that supplementing ammonium besides glucose did not fully restore the loss of symbiont density in *Aiptasia* and *Cassiopeia* (Figure S1) further suggests that glucose may have additional effect(s) to the symbiotic relationships.

We sincerely appreciate the reviewer's insightful comments and suggestions. We have thoroughly considered each point raised and have made the necessary revisions to address them. Our detailed responses to each concern are provided below:

1) We acknowledge the reviewer's concern regarding the measurement of GS/GOGAT activity and its relevance to our hypothesis. Firstly, we apologize for the confusion regarding the formulation of our hypothesis. We appreciate the clarification provided by the reviewer. We now emphasize that our hypothesis was developed based on a meta-analysis of transcriptomic studies and ¹³C bicarbonate labeling experiments (Cui et al., 2019, PLoS Genetics). Additionally, we have conducted experiments to measure the activities of GS and GOGAT in response to nutrient supplementation (lines 164 – 169, 481 – 492). The new data and corresponding analysis have been included in Figure S2. These measurements support our hypothesis by demonstrating that host GS and GOGAT are actually active in all three species and that glucose or glucose + ammonium supplementation generally enhances their activities, while ammonium supplementation alone suppresses their activities. These findings provide further evidence for the incorporation of C and N into host metabolites via the host GS/GOGAT cycle, reinforcing the proposed negative feedback response driven by glucose and ammonium availability.

2) We agree with the reviewer that glucose can act as a signaling molecule in various processes, in addition to its nutritional role. However, our experiments do not support the hypothesis that glucose triggers dysbiosis and bleaching. We observed that the animals treated with glucose remained healthy without exhibiting signs of stress. Furthermore, transcriptomic analysis comparing glucose-treated symbiotic and aposymbiotic anemones with non-treated ones revealed no significant changes in genes involved in stress responses (see also new Figure 3; lines 147 – 163, 171 – 180, and 459 – 480). Instead, the differentially expressed genes were primarily enriched biological processes and pathways related to amino acid biosynthesis. We have expanded our discussion in the revised manuscript, emphasizing that the carbon-nitrogen balance likely regulates the symbiotic relationship at a generalized top level. We acknowledge that further investigations are required to elucidate the detailed molecular mechanisms downstream of the metabolic responses and to differentiate the changes in molecular signaling from the response of different metabolic states.

3) We appreciate the reviewer's observation regarding the incomplete restoration of symbiont density when ammonium was supplemented alongside glucose in *E. diaphana* and *C. andromeda*. We would like to clarify that the same ammonium concentration was used for all three species to facilitate comparison. However, given the different capacities of these species in regulating the carbon-nitrogen balance, as indicated by the enzyme activity assays and isotope labeling experiments, we expected variations in the response to the same ammonium concentration. To address the reviewer's concern, we now conducted additional supplementation experiments using different concentrations of ammonium in *E. diaphana* and *C. andromeda*. The results (Figure S1; lines 119 – 134) demonstrate that increasing ammonium concentration increases symbiont density and eventually restores it to the control level in both species. These findings support our hypothesis and confirm that the symbiont population change is indeed dose-dependent as predicted by the model. Furthermore, the experiments further confirm the higher capacity for ammonium assimilation in *E. diaphana* and *C. andromeda* compared to *S. pistillata*, as they require nearly twice as much ammonium to reach control symbiont densities. These new results provide additional support for our hypothesis and underscore the potential differential sensitivities of these species to environmental perturbations.

We sincerely thank the reviewer for their valuable input, which has significantly improved the clarity and strength of our study. We have incorporated the measurements of GS/GOGAT activity, addressed the alternative hypothesis, and provided further evidence for the species-specific responses to glucose and ammonium supplementation.

In addition, here are some more comments to further improve the manuscript.

Figure 2 showed the symbiont cell density changes induced by the availability of glucose and ammonium. "Greek letters indicate statistical differences with a significance cut-off at $p < 0.05$." - which samples were conducted for the statistical test?

Thank you for your question. We apologize for any confusion caused by the statement in Figure 2. To clarify, the assignment of Greek letters in Figure 2 was based on pairwise comparisons that encompassed all possible pairs across the different conditions tested. Conditions with different Greek letters indicate a statistically significant difference ($p < 0.05$) between them. On the other hand, conditions sharing the same Greek letter indicate that their pairwise comparison yielded a p-value greater than 0.05, signifying no statistical difference between them. We have revised the figure caption (lines 141–142) to provide a clearer explanation of the statistical analysis conducted.

Figure S1 showed the combined effects of glucose and ammonium on symbiont cell density changes. Was there any particular reason to separate this data from Figure 2? Logically it should be included in Figure 2. One thing to note is that the symbiont density in the control samples was a little bit different for *Aiptasia* and coral in those two figures.

Thank you for bringing up the differences in the control samples between Figure 2 and the previous Figure S1. We appreciate your observation. The reason for separating the data into different figures was that the experiments shown in these two figures were conducted independently using different batches of samples. Therefore, we presented them separately to maintain experimental integrity and clarity in the results.

Regarding the slight differences in symbiont density observed in the control samples of *E. diaphana* and *S. pistillata* between the two figures, it is important to note that biological replicates can exhibit natural variation. Factors such as differences in feeding regimen, light intensity, and water quality, among others, can contribute to variations in symbiont density even within the same species. We acknowledge that these slight differences may exist and have mentioned them in the manuscript (lines 398 – 400) to ensure transparency in our experimental procedures.

Figure 3: It would be beneficial to label the metabolites clearly in the figure... and in its current form this figure is confusing... While obviously ¹³C-glucose and ¹⁵N-ammonium were incorporated into the host metabolites, which is not surprising, it is unclear how it is linked to regulating symbiont density. The authors may want to clarify it in the results and discussion.

We appreciate the reviewer's suggestion to label the metabolites clearly in Figure 3 (now Figure 4). We have made the necessary revisions by reorganizing the figure legend and adding a clarification below the figure (lines 215 – 216). The labels now correspond to the same order as presented in the figure legend, making it easier to understand and follow.

Furthermore, we would like to express our gratitude for the comment regarding the clarification of the connections between host-dependent amino acid synthesis and symbiont density regulation. We have taken this suggestion into account and have expanded our explanations throughout the Results and Discussion sections, as shown in lines 118 – 133, lines 183 – 188, lines 286 – 306.

Line 190 and Figure 4: The authors mentioned “host-driven ammonium assimilation”, however, it should be pointed out that the symbiont algae can also assimilate nitrogen and provide amino acids to the host. In particular, figure 4 showed the differences in the incorporation of ¹³C between symbiotic and symbiont-free animals, and including the role of the algal symbiont would enrich the discussion.

We appreciate the reviewer's comment regarding the assimilation of ammonium by the symbiont algae. We have taken this into consideration and expanded our discussion on the metabolic competition between symbiont-dependent and host-driven ammonium assimilation (lines 342 – 351).

While it is indeed challenging to separate the contributions of nitrogen assimilation between the host and symbiont, we acknowledge that both partners have the ability to assimilate nitrogen for their own benefit. In a nitrogen-limited environment, both the host and the symbiont compete to assimilate and retain nitrogen for their respective growth. Considering that the symbionts are nitrogen-limited in symbiosis, it can be assumed that they do not provide significant amounts of nitrogenous products to the host, as otherwise, they would not be nitrogen limited.

Regarding Figure 4 (now Figure 5), which illustrates the differences in the incorporation of ¹³C between symbiotic and symbiont-free animals, we agree that it would be beneficial to discuss the role of the algal symbiont in more detail. The observed differences in ¹³C incorporation likely reflect the mixed utilization of externally supplied ¹³C-labeled glucose and symbiont-derived ¹³C-free glucose in symbiotic anemones. Consequently, the symbiotic animals exhibit slightly lower ¹³C incorporation compared to aposymbiotic anemones, as they have both sources of carbon backbones available to them.

We have incorporated these points into our manuscript (lines 313 – 315, 342 – 357) to provide a more comprehensive understanding of the metabolic competition and the contributions of both the host and symbiont in nitrogen assimilation.

Line 262: “our findings suggest that cnidarian hosts rely rather heavily on the provision of glucose to control their symbiont population”, again, alternative hypotheses, for example, the signaling role of glucose, would be beneficial for interpreting and discussing the results.

We appreciate the reviewer's suggestion and agree that alternative hypotheses should be considered for interpreting and discussing the results. Our new experiments now better demonstrate that symbiont density is controlled by the balance between the availability of glucose and ammonium (lines 119 – 134). Specifically, in *E. diaphana* and *C. andromeda*, increasing the concentration of ammonium to double the amount found in *S. pistillata* effectively restored the symbiont density to control levels. The new transcriptomic analysis (lines 143 – 163) further supports our findings by revealing glucose-induced amino acid biosynthesis. Additionally, the UHPLC-HR-MS data demonstrate the co-incorporation of carbon and nitrogen into host metabolites. Collectively, these findings provide compelling evidence for our model.

However, we would like to emphasize that our focus in this study was on the metabolic regulation of symbiont density by the availability of glucose and ammonium. While our results support the role of glucose as a metabolic regulator, we acknowledge that glucose or its derivatives may also have signaling effects that play critical roles in regulating symbiosis at different levels beyond symbiont density. We recognize the importance of considering alternative hypotheses, including the potential signaling role of glucose, and we have now expanded our discussion (lines 286 – 306, 334 – 337, and 342 – 357) to provide a more comprehensive picture.

Reviewer #2 (Remarks to the Author):

The manuscript uses well-controlled stable isotope labelling to explore the mechanism(s) of symbiont population control and organic C and inorganic N assimilation in the cnidarian-dinoflagellate symbiosis. This is important work both for advancing symbiotic theory but also in the contexts of holobiont stability in response to environmental perturbation (e.g. coral bleaching). The work is methodologically sound and well performed, and the manuscript well written. My concerns are with the novelty and interpretation of the dataset, and the paper may be acceptable after addressing the following questions:

Please state clearly the novelty of this dataset relative to a variety of papers investigating C/N uptake dating back to the 1980s. The hypothesis tested, nitrogen as limiting to symbiont population growth, and organic C supplementation resulting in large declines in symbiont population is not novel. This work tests it elegantly and with more breadth and resolution (the MS work is strong!), but the novelty must be very clearly stated to differentiate it from past work.

We appreciate the reviewer's comment and agree that it is important to clearly state the novelty of our dataset relative to previous studies investigating C/N uptake in the cnidarian-dinoflagellate symbiosis. While nitrogen limitation and the impact of organic carbon supplementation on symbiont population have been reported in previous studies, our work provides several novel contributions.

First, our study connects these two aspects as part of the same regulatory mechanism, providing a comprehensive model for symbiont control in response to the availability of organic carbon. This model, which proposes a negative feedback loop between glucose availability and symbiont density, has not been proposed in this form previously. To our knowledge, we were the first to propose it in Cui et al 2019, PLoS Genetics.

Second, we are the first to offer an experimental demonstration of this model as well as a more detailed and comprehensive examination of the underlying molecular pathways and enzymes involved in this mechanism.

Third, our study extends the investigation to three cnidarian species from different classes, namely *E. diaphana*, *C. andromeda*, and *S. pistillata*. By demonstrating the universality of this mechanism across different cnidarian hosts that evolved symbiosis independently, we provide evidence for its evolutionary significance and broad applicability.

Lastly, the use of stable isotope labeling and advanced mass spectrometry techniques, as highlighted by the reviewer, contributes to the resolution and depth of our study, providing detailed insights into the pathways underlying the incorporation of carbon and nitrogen into host metabolites.

In summary, while some aspects of our findings have been reported in previous studies, the novelty of our study lies in the connection of previous findings and their incorporation into the model we propose, the experimental demonstration, the detailed examination of molecular pathways and enzymes, the extension to multiple cnidarian species, and the use of advanced techniques for data resolution. We now ensure that these points are clearly stated in the manuscript (lines 87 – 94, 148 – 152, 183 – 186, 237 – 239, 284 – 306, and 352 – 357) to differentiate our work from previous studies and highlight its novelty and contributions to the field.

My understanding was that ammonium assimilation in the coral symbiosis was primarily via GS/GOGAT in the symbionts, not the host (see Pernice et al 2012 ISME, A single-cell view...). This is mentioned only in passing. How does algal ammonium uptake (and potential diversity in ammonium affinity amongst symbiont species) affect your model?

We appreciate the reviewer's feedback and clarifications on the topic of ammonium assimilation in coral symbiosis. You are correct that Pernice et al. (2012) demonstrated higher per-biomass nitrogen assimilation rates in the symbionts compared to the coral host. However, this study did not consider the higher total biomass of the host. When considering the overall biomass of the host, recent research, including our own work (Cui et al. 2023 Sci Adv), indicates that the host can assimilate a comparable amount of ammonium to the symbionts. We apologize for not explaining this distinction clearly in the manuscript.

In our model, we propose that the availability of organic carbon (e.g., glucose) regulates the symbiont population by controlling the host-driven ammonium assimilation. This regulatory mechanism ensures a balance between nitrogen availability and symbiont proliferation. The symbionts, indeed, utilize glutamine synthetase/glutamate synthase (GS/GOGAT) to assimilate ammonium. The competition for ammonium uptake and assimilation between the host and symbionts is a critical aspect of the model.

Regarding the potential diversity in ammonium affinity among different symbiont species, it is an important consideration. Symbiont strains with higher ammonium assimilation rates may outcompete hosts with lower ammonium assimilation capacity, leading to imbalances in nutrient exchange. On the other hand, symbiont strains with lower ammonium assimilation capacity may struggle to colonize hosts with high ammonium assimilation capacity. The capacity of both the host and symbiont to assimilate ammonium could therefore affect the viability of host-symbiont combinations and thus contribute to host-symbiont specificity.

We appreciate the reviewer highlighting these aspects. We have revised the manuscript (lines 286 – 306, 313 – 315, and 342 – 357) to provide a better explanation of the contribution of the host and symbiont to holobiont ammonium recycling and the potential impact of symbiont diversity on our model and host-symbiont specificity.

The supposition on different degrees of ¹³C glucose incorporation in different taxa (e.g. line 227 forward) is weakened by only having 1-2 members of each taxon, limiting the ability to interpret the data. I noticed replication in another coral, *Acropora*, in the supplemental data (Fig S2). This should absolutely be moved into the main text and figures to strengthen the manuscript. However, it should still be noted that there are not enough representative taxa (n = 1-2) to be conclusive about entire subphyla. I would refer to the organisms by the genus name throughout the manuscript, rather than “anemones” or “jellyfish” as it implies that these patterns would necessarily hold throughout the groups.

We appreciate the reviewer's feedback regarding the representation of taxa and the terminology used in the manuscript. You raise a valid point that having 1-2 members of each taxon is not sufficient to draw conclusions about entire subphyla. We agree with this perspective. While it would be less parsimonious to assume that members of the same subphylum evolved different mechanisms, we acknowledge the limitations of our sample size.

In response to your suggestion, we have changed the terminology in the manuscript to refer to the specific organisms by their genus names, such as “*E. diaphana*” and “*C. andromeda*,” or by explicitly mentioning “anemone *E. diaphana*” and “jellyfish *C. andromeda*.” This modification provides a clearer representation of the specific organisms studied and avoids generalizations that may not hold throughout the entire subphyla.

Regarding the inclusion of *A. hemprichii* in the main text and figures, we have now incorporated the data into the main figures to strengthen the manuscript. Specifically, we have moved the relevant figure (Fig S2) to the main Figure 2, allowing for a more comprehensive representation of the findings across different coral taxa.

We appreciate the reviewer's valuable input, which has contributed to improving the clarity and presentation of our research.

Ln174: “To further determine if the observed assimilation of ammonium is driven by the host animals...” I believe here you've shown that the (aposymbiotic) host can assimilate ammonium, especially when supplemented with glucose, but that doesn't mean that much ammonium is assimilated by the host when in symbiosis, as opposed to by the symbionts. Again, how does algal assimilation of ammonium complicate your analysis? Could the ¹⁵N you're detecting arise from algal assimilation and translocation of N-containing compounds to the host (amino acids etc.) and/or secondary assimilation of after catabolism of those compounds?

We appreciate the reviewer's comments and the opportunity to clarify our findings. As previously mentioned, we developed our model based on observed transcriptomic changes that demonstrated the activation of the GS/GOGAT pathway and amino acid biosynthesis in *E. diaphana* in response to symbiosis. Therefore, our model is not based on observed changes that happen in aposymbiotic anemones in response to glucose supplementation but on the changes observed in anemones in response to becoming symbiotic. Subsequent experiments, including our previous study in Cui et al 2019 as well as the metabolomic data, new RNA-seq data, and enzyme activity assays, provided additional experimental evidence supporting the general finding that symbiotic anemones assimilate ammonium using glucose as a carbon backbone.

While we acknowledge that the algal assimilation of ammonium complicates the analysis, it is important to consider that the symbionts are nitrogen limited. In a nitrogen-limited condition, it is challenging to understand how the symbionts could provide significant amounts of nitrogenous products, such as amino acids, to the host. The limited availability of nitrogen suggests that the symbionts are primarily competing with the host for ammonium assimilation. If the symbionts did indeed assimilate the ammonium and translocate it back to the host, they would obviously not be nitrogen limited. The provision of additional ammonium would then not result in increased symbiont densities but in increased amino acid production and translocation to the host. However, this is not what we and many others have observed. Instead, the provision of ammonium results in increased symbiont densities which clearly proves that most of the assimilated nitrogen is used for cell proliferation.

We recognize that our model simplifies a complex metabolic interaction, and there may be additional factors at play. However, the evidence we have presented strongly supports the role of the host in driving ammonium assimilation through the activation of the GS/GOGAT pathway and amino acid biosynthesis. This model provides a framework for understanding the regulation of symbiont density in response to glucose availability.

More minor concerns:

- The title of the paper is somewhat misleading, in that it implies that this is an evolutionary study when it is not.

Thanks for the comment. While our study is not directly an evolutionary study, it contributes to our understanding of the mechanisms underlying the repeated evolution of these associations. The identification of a shared regulatory mechanism in three independently evolved symbioses suggests a convergent evolutionary pattern driven by the host's ability to control symbiont populations via GS/GOGAT-mediated amino acid biosynthesis. We, therefore, believe that the current title is the best option to convey the message.

- Ln233: "This difference in the incorporation of ^{13}C might result from physiological differences in carbon requirements and utilization..." This seems vague, and I'm not sure how to interpret it.

We apologize for the confusion caused. We wanted to say that the differences we observed in different animals might result from the differences in their nitrogen assimilation capacities. This has been shown in the new enzyme activity assays. But this does not rule out the possibility that the symbionts they harbor may also play critical roles.

- Ln310: What is meant by "a jointed experiment with four treatments in parallel"? Are these data meant for another manuscript?

We apologize for the confusion caused. The sentence refers to the results shown in the previous Figure S2. We have rephrased this section (lines 393 – 402) to make it clearer and cited the corresponding figures to avoid confusion.

- Fig. 3 is elaborate and I appreciate the amount of effort that went into it, but I find the heat maps to be quite unintuitive. Is there a more direct way to display these data?

We apologize for any confusion caused by the heat maps in Figure 3 (now Figure 4). We appreciate your feedback and understand that alternative visualizations may be more intuitive for interpreting the data.

During the process of creating Figure 3, we explored various ways to represent the complex dataset, including bar charts and dot plots. However, we found that these alternative visualizations resulted in a cluttered and less interpretable representation of the data.

In order to strike a balance between complexity and clarity, we opted to use heat maps as they allow us to display multiple variables simultaneously and provide a comprehensive overview of the data. Heat maps also enable the visualization of patterns and trends across different conditions and treatments.

To assist with interpretation, we have included supplementary tables (Table S5) that present the numerical values corresponding to the heat map data. This additional information can be used to further analyze and understand the specific values and patterns depicted in the heat maps.

We understand that different individuals may have varying preferences when it comes to data visualization. We appreciate your feedback and will take it into consideration for future presentations of similar datasets.

Reviewer #3 (Remarks to the Author):

The manuscript by Cui et al. describes an elegant study that provides experimental evidence for the hypothesis that a negative carbon-nitrogen feedback loop is responsible for tightly controlled symbiont cell number in a variety of cnidarian host species. To this end, the authors tested the predictions of the model in three different cnidarian model systems that independently evolved symbiosis with Symbiodiniaceae. First, they administered exogenous glucose and ammonium to the animals and observed the numbers of the symbionts. These experimental data were supplemented by extensive stable isotope analyses. These data support the negative feedback model based on nutrient flux.

We thank the reviewer for the positive comments and constructive suggestions. We have conducted additional experiments as suggested and made the corresponding changes in the manuscript. Our point-by-point responses are as follows.

Specific comments:

Line 57: Please explain GS/GOGAT.

Thanks for pointing this out. We have included the full name of GS and GOGAT now.

"... symbiosis activates **glutamine synthetase / glutamate synthase (GS/GOGAT)** mediated amino acid biosynthesis..."

Figure 1: I think, it would be helpful to include symbiont proliferation to the model, as it is your read out in your experiments.

Thanks for the suggestion. We have re-phrased the caption of Figure 1 (lines 73 – 85) to clarify symbiont proliferation changes at different symbiotic stages and in response to differences in nitrogen availability.

Figure 2. Please provide in the legend the concentrations of ammonium and glucose used.

Thanks for the suggestion. We have included the concentrations in the figure caption now to avoid crowded texts in Figure 2, d–g.

Figure 2: To show these effects more convincingly, I propose to show that the effects of glucose and ammonium on symbiont cell numbers are concentration dependent as predicted; at least for one model species.

Thanks for this constructive suggestion. We have performed the suggested experiments and included the results in the revised Figure 2.

Figure 2: Why is the number of symbiont cells normalized to μg of protein rather than host cells? Based on your model, I would guess that protein content is not a good choice for normalization because AS synthesis is not independent in your experiment. It is directly dependent on the amount of glucose and ammonium provided. Please provide additional evidence of the changed symbiont cell numbers, e.g., by microscopic analyses of individual host cells.

Thank you for your question and suggestions regarding Figure 2. We appreciate your insights and the considerations you raised.

You are correct that protein content may not be an independent measure for normalizing symbiont cell numbers, as the synthesis of amino acids in the host is directly dependent on the availability of glucose and ammonium. We acknowledge this potential limitation and have taken additional steps to address it.

In our study, we performed several experiments to evaluate different normalization methods for quantifying symbiont cell numbers. We explored normalizing coral fragment color changes to the color scheme presented in the Coral Health Chart (CoralWatch), as well as normalizing symbiont numbers to the surface area of sea anemone tentacles. These alternative methods provided consistent results with respect to the impact of glucose and ammonium on symbiont density, supporting the findings obtained through protein content normalization. We have included these experiments in Figure S27 and provided a description of the methods in the manuscript (lines 446 – 460).

While these alternative methods yielded similar patterns, they were time-consuming, difficult to scale up for high-throughput analyses, and not easily comparable across species. As a result, we decided to utilize protein content as the normalization method for symbiont cell numbers in Figure 2.

We acknowledge the potential limitations of this normalization approach and the dependency of symbiont density on amino acid synthesis, which is influenced by glucose and ammonium availability. However, by comparing multiple normalization methods and obtaining consistent results, we gained confidence in the findings and conclusions of our study. Future research could explore alternative normalization approaches or additional analyses, such as microscopic examination of individual host cells, to provide further evidence and support for the changes in symbiont cell numbers.

Thank you for highlighting this aspect, and we have revised the manuscript to provide a clearer explanation of our normalization approach and the associated considerations.

Line 108; Figure S1: Please be precise here, in *E. diaphana* the effect was significant.

Thanks for pointing this out. We have made the changes accordingly.

Line 293: Please describe the exact conditions for the treatment, timing, light conditions, feeding, etc.

Thanks for the suggestion. We have revised the corresponding method section (lines 389 – 402) to make it clearer.

Line 306: Please show the data in the supplements.

Thanks for the suggestion. We have modified this section according to comments from other reviewers and included the data in Figure 2.

Discussion: Given the effects of glucose and ammonium on symbiont cell number, it would be relevant to discuss the mechanisms and timing of cell number adjustment. How quickly do the changes occur, especially the reduction in symbiont cells? Do symbiont cells die or are they just not proliferating? How long does a symbiont cell live? Do they get expelled from the host? I would encourage the author to discuss these points in more detail.

Thank you for the valuable comment and suggestion. We appreciate your insights into the mechanisms and timing of symbiont cell number adjustment. We have incorporated your suggestions into our discussion (lines 286 – 306, 334 – 337) to provide a more detailed understanding of these aspects.

Based on our findings, we propose that symbiont population adjustment is regulated by different processes. First, by the availability of nitrogen which regulates proliferation rates. As soon as proliferation rates are lower than the rates of decay (i.e. rate of symbiont death or expulsion) the symbiont population slowly decreases. However, we also observe much faster reductions in symbiont density, specifically in corals. We hypothesize that the host animal might be able to sense changes in its own metabolic status. When there is a sufficiently high of glucose available, the host animal initiates the expulsion of no-longer-needed symbionts.

These insights into the mechanisms and timing of symbiont cell number adjustment provide a more comprehensive understanding of the host-symbiont interactions and the regulatory processes involved. We have revised the manuscript to incorporate this discussion and highlight the relevance of these points. Thank you for bringing these aspects to our attention.

REVIEWER COMMENTS

Reviewer #1 (Remarks to the Author):

The revised manuscript incorporates new data of transcriptomics analysis in response to glucose supplementation, which showed differentially expressed genes involved in signal transduction, such as Wnt signaling. This suggests that the observed effects might extend beyond merely metabolic functions, highlighting the possible role of signal transduction during glucose supplementation and underscoring the need for careful interpretation of its effects on symbiotic systems. While recent findings highlighted the role of glutamine synthetase / glutamate synthase (GS/GOGAT) mediated amino acid biosynthesis activated by symbiosis, the novelty of this study seems limited since earlier studies drew similar conclusions using comparable methods (a majority of the Response to Reviewer Comments were focused on prior findings...). Figure 2 seems to conglomerate plots from supplemental figures, incorporating data into a single plot for each species would enhance clarity. The interpretation of Figure 4 needs further clarity regarding the proportion of metabolites incorporated from photosynthetically derived carbon in the symbiotic animals.

Please see below the detailed comments:

Line 55-59: “ A recent meta-analysis of transcriptomic data comparing symbiotic and aposymbiotic *Exaiptasia diaphana* revealed that symbiosis activates glutamine synthetase / glutamate synthase (GS/GOGAT) mediated amino acid biosynthesis in the host. Metabolomic analyses of intermediate metabolites further confirmed that symbiotic anemones use symbiont-derived photosynthates to assimilate waste ammonium”. It still appeared to be a limited novelty of this study as previous studies have essentially conveyed the same conclusion using similar analytical methods....

Figure 2: plots from previous supplemental figures squeezed together in this figure. Figure 2g is a good example of clearly showing the impact of ammonium, glucose, and both within the same plot. I understand that treatment and experiments were performed in different batches, but it would be nice if the data from other species were also shown in one plot, not separately in different plots.

Figure 2: in *Aiptasia* the symbiont density reduction by glucose seemed to saturate around 10 mM glucose, as there were little differences even up to 100 mM glucose.

Figure S2: what were the concentrations of glucose or ammonium used? It would be helpful to include the information in the legend.

Line 166-168: “We found that both host GS and GOGAT showed activity in all three species and treatments, and that glucose generally promoted the activity of both enzymes while ammonium reduced it (Figure S2, a–f).” I would recommend conducting statistical analysis and describing the data more accurately as this study compared the responses from *Aiptasia*, *Cassiopea*, and coral.

Line 168-169: “The overall pattern of enzyme activity changes aligned surprisingly well with the pattern we observed in glucose/ammonium-induced symbiont density changes.”. Again a more accurate description of the figure would be recommended. Also it may be helpful if the authors clarify why the data was “surprisingly”.

Figure 3: transcriptome analysis showed differentially expressed genes involved in a few categories including splicing, ribonucleoprotein complex, amino acid biosynthesis, as well as wnt signal transduction, which begin with proteins that pass signals into a cell through cell surface receptors. These data suggest that signal transduction may be involved during glucose supplementation, besides activation of GS/GOGAT. This seemed to be consistent with the fact the symbiont density reduction saturates around 10 mM glucose. The authors would need to be careful drawing the conclusions about the impact of glucose to the symbioses systems.

Line 205: not just *Aiptasia*

Figure 4: were the scales $\log_2(\text{fold change})$ in the heatmaps? This needs to be clarified.

Line 231-233: “ In particular, most of the ^{15}N isotope was identified in both ^{13}C - and ^{15}N -containing metabolites ($^{13}\text{C}^{15}\text{N}$) from animals with the combined treatment, while only a small proportion of the ^{15}N isotope ended up in $^{12}\text{C}^{15}\text{N}$ compounds ”. This data appeared

to confirm that the supplemented glucose and ammonium were metabolized into ^{13}C - and ^{15}N -containing metabolites. In contrast, only a small portion of ^{12}C , which was potentially from algal photosynthesis, appeared to be incorporated into $^{12}\text{C}^{15}\text{N}$ compounds... There was some accumulation of $^{12}\text{C}^{15}\text{N}$ metabolites under the condition supplemented with ^{15}N , but similar in Aiptasia-Apo... So it was unclear about the incorporation of the photosynthetically derived carbon. In any case, the authors may need to clarify the data interpretation from this experimental design...

Reviewer #2 (Remarks to the Author):

The authors have adequately addressed my concerns with the manuscript.

Reviewer #3 (Remarks to the Author):

I have no further comments on the manuscript. All my points are answered comprehensively and satisfactorily.

REVIEWER COMMENTS

Reviewer #1 (Remarks to the Author):

The revised manuscript incorporates new data of transcriptomics analysis in response to glucose supplementation, which showed differentially expressed genes involved in signal transduction, such as Wnt signaling. This suggests that the observed effects might extend beyond merely metabolic functions, highlighting the possible role of signal transduction during glucose supplementation and underscoring the need for careful interpretation of its effects on symbiotic systems. While recent findings highlighted the role of glutamine synthetase / glutamate synthase (GS/GOGAT) mediated amino acid biosynthesis activated by symbiosis, the novelty of this study seems limited since earlier studies drew similar conclusions using comparable methods (a majority of the Response to Reviewer Comments were focused on prior findings...). Figure 2 seems to conglomerate plots from supplemental figures, incorporating data into a single plot for each species would enhance clarity. The interpretation of Figure 4 needs further clarity regarding the proportion of metabolites incorporated from photosynthetically derived carbon in the symbiotic animals.

We would like to thank the reviewer for the comments and suggestions. In the following point-by-point responses, we have addressed all the comments regarding the interpretation of Wnt signaling, previous studies of GS/GOGAT, the layout of Figure 2, and the interpretation of Figure 4. Having said so, we would like to start by responding to the comment regarding the novelty of our study.

The reviewer mentions that the potential role of GS/GOGAT-mediated amino acid synthesis as a mechanism for symbiont control has been highlighted in previous studies, which limits the novelty of our study. In response to this statement, we would like to point out that we specifically emphasized these previous findings in response to the previous comments of reviewers 1 and 2 regarding the possibility that host responses might be different in symbiosis. Our aim was to clarify that we based our metabolic model specifically on the observed host responses to symbiosis. We believe that providing this background information does not diminish the novelty of our study, which actually focuses on testing the metabolic model, i.e., showing that symbiont density is really controlled through the balance of carbon translocation and nitrogen availability, which has not been tested in any previous study.

As mentioned in our previous revision, we would like to reiterate the unique contributions of our research:

First, our study connects nitrogen limitation and the impact of organic carbon supplementation on symbiont population as part of the same regulatory mechanism. We thus provide a comprehensive model for symbiont control in response to the availability of organic carbon. This model, which proposes a negative feedback loop between glucose availability and symbiont density, has not been proposed in this form previously.

Second, we are the first to offer experimental demonstrations of this model as well as a more detailed and comprehensive examination of the underlying molecular pathways and enzymes involved in this mechanism.

Third, our study extends the investigation to three cnidarian species from different classes, namely *E. diaphana*, *C. andromeda*, and *S. pistillata*. By demonstrating the universality of this mechanism across different cnidarian hosts that evolved symbiosis independently, we provide evidence for its evolutionary significance and broad applicability.

Lastly, the use of stable isotope labeling and advanced mass spectrometry techniques, as highlighted by another reviewer, contributes to the resolution and depth of our study. It further provides detailed insights into the pathways underlying the incorporation of carbon and nitrogen into host metabolites and is the first study to provide relative isotope incorporation rates into the intermediate metabolites of the proposed molecular pathway.

In summary, our work builds on prior findings, but it's the first to provide experimental proof for the proposed metabolic model and its experimental validation in multiple species that evolved Symbiodiniaceae symbioses independently.

Please see below the detailed comments:

Line 55-59: “ A recent meta-analysis of transcriptomic data comparing symbiotic and aposymbiotic *Exaiptasia diaphana* revealed that symbiosis activates glutamine synthetase / glutamate synthase (GS/GOGAT) mediated amino acid biosynthesis in the host. Metabolomic analyses of intermediate metabolites further confirmed that symbiotic anemones use symbiont-derived photosynthates to assimilate waste ammonium”. It still appeared to be a limited novelty of this study as previous studies have essentially conveyed the same conclusion using similar analytical methods....

We would like to thank the reviewer for drawing attention to the overlap with previous works in lines 55-59. However, as mentioned in our previous response, this finding does not constitute the major finding of our study. While “previous studies have essentially conveyed the same conclusion using similar analytical methods” regarding the putative involvement of GS/GOGAT mediated amino acid biosynthesis in symbiont control, they did not provide actual experimental proof. Furthermore, these studies did not test if this mechanism of control is indeed dependent on the availability of glucose as proposed, which is an essential part of the metabolic model. Finally, while previous studies centered on sea anemones, we expanded our scope, aiming to generalize findings across Cnidaria.

However, to avoid confusion, we have revised the corresponding texts:

“A recent meta-analysis of transcriptomic data comparing symbiotic and aposymbiotic *Exaiptasia diaphana* revealed that symbiosis activates glutamine synthetase / glutamate synthase (GS/GOGAT) mediated amino acid biosynthesis in the host¹¹. Metabolomic analyses further confirmed that symbiotic anemones increase waste ammonium assimilation, likely due to the availability of symbiont-derived photosynthates^{11,17}.”

Figure 2: plots from previous supplemental figures squeezed together in this figure. Figure 2g is a good example of clearly showing the impact of ammonium, glucose, and both within the same plot. I understand that treatment and experiments were performed in different batches, but it would be nice if the data from other species were also shown in one plot, not separately in different plots.

We appreciate and understand the reviewer's comment. The figures 2d-g were initially listed as supplementary figures. We were asked to move them into the main figures since they represent important information regarding the general pattern of symbiont population regulation in response to different nutrient supplementations. However, we are afraid that these figures cannot be further integrated with figures 2a-c since the data were obtained from different batches of experiments. Having said so, we believe that the pattern is clear and easy to understand with the current layout.

Figure 2: in *Aiptasia* the symbiont density reduction by glucose seemed to saturate around 10 mM glucose, as there were little differences even up to 100 mM glucose.

We agree with the reviewer's interpretation. Increasing glucose concentration above 10 mM can still trigger further decreases in symbiont density, but the effects are marginal compared to 10 mM glucose. These results suggest that *Exaiptasia* is very efficient in assimilating waste ammonium even at lower glucose concentrations, which is consistent with our other findings.

Figure S2: what were the concentrations of glucose or ammonium used? It would be helpful to include the information in the legend.

We appreciate the reviewer's suggestion and have modified the figure to include the information on concentrations used in the experiments.

Line 166-168: "We found that both host GS and GOGAT showed activity in all three species and treatments, and that glucose generally promoted the activity of both enzymes while ammonium reduced it (Figure S2, a-f)." I would recommend conducting statistical analysis and describing the data more accurately as this study compared the responses from *Aiptasia*, *Cassiopea*, and coral.

We appreciate the reviewer's suggestion and have included the statistical test information in the corresponding Figure S2. However, we would like to note here that all experiments were performed in symbiotic individuals which have access to symbiont photosynthates. Therefore, the activity of these enzymes can already be expected to be increased so that treatments have only a reduced effect on activity changes.

Line 168-169: "The overall pattern of enzyme activity changes aligned surprisingly well with the pattern we observed in glucose/ammonium-induced symbiont density changes." Again a more accurate description of the figure would be recommended. Also it may be helpful if the authors clarify why the data was "surprisingly".

We thank the reviewer for pointing out the need for clarification on lines 168-169. In our observations, we identified consistent patterns across different species: glucose generally enhances GS/GOGAT activity while decreasing symbiont density, whereas ammonium tends to suppress enzyme activities and elevate symbiont density. Given that enzyme activity assays on total cell lysates often do not display distinct patterns, these consistent findings across species were noteworthy to us. However, we acknowledge that the term "surprisingly" may not convey our intent adequately in a scientific context. As such, we have opted to remove it from the manuscript for clarity.

"The overall pattern of enzyme activity changes aligned well with the pattern observed in glucose/ammonium-induced symbiont density changes: glucose generally enhances GS/GOGAT activity while decreasing symbiont density, whereas ammonium tends to suppress enzyme activities and elevate symbiont density."

Figure 3: transcriptome analysis showed differentially expressed genes involved in a few categories including splicing, ribonucleoprotein complex, amino acid biosynthesis, as well as wnt signal transduction, which begin with proteins that pass signals into a cell through cell surface receptors. These data suggest that signal transduction may be involved during glucose supplementation, besides activation of GS/GOGAT. This seemed to be consistent with the fact the symbiont density reduction saturates around 10 mM glucose. The authors would need to be careful drawing the conclusions about the impact of glucose to the symbioses systems.

We thank the reviewer for highlighting the nuances of Wnt signaling from Figure 3 and its potential implications during glucose supplementation. However, we would like to point out that this was a hypothesis-driven study, meaning that we formulated a hypothesis and designed experiments to test it. We specifically added the gene expression data to show that the expression of GS/GOGAT components is induced by symbiosis and that the corresponding genes responded as predicted by our model. As for the comment that we cannot exclude other potential signaling functions of glucose, we fully agree with the reviewer, but it was also not our intention to exclude other functions. It is, in fact, nearly impossible to prove that something does not exist, or as in this case, that glucose signaling is not involved in the observed responses. Therefore, we have designed this study to test our hypothesis, and we provide the results of our experiments, which provide direct evidence for our metabolic model and hypothesis.

Having said that, we tried to explore potential signaling effects of glucose that could lead to dysbiosis and the associated decrease in the symbiont population, as suggested by the reviewer. Despite our efforts, we did not find any convincing evidence of signaling pathways causing dysbiosis in our RNA-

seq data. Rather, the data consistently highlighted amino acid biosynthesis pathways, particularly those associated with GS/GOGAT, reinforcing our carbon-nitrogen negative feedback model.

We agree with the reviewer regarding the generalized operation of signaling transductions, where extracellular signals are transmitted across the cell membrane. However, we emphasize that distinct pathways have unique roles. The canonical Wnt signaling pathway we observed in our data is well-known for its role in regulating cell proliferation. The upregulation of these genes makes sense in our glucose-supplementation experiment. With more nutrients available (in this case, glucose), an uptick in cell proliferation and the associated pathways (transcription, translation, as well as Wnt pathway) is expected.

Moreover, symbiont density reduction was found to align with the downregulation of Wnt signaling genes (as highlighted by Maor-Landaw et al., 2014). This is different from the general upregulation we see in our glucose supplementation experiments. This further supports our stance that the observed reduction in symbionts due to glucose doesn't tie back to the upregulation of the Wnt signaling pathway.

Regarding the signaling effect and the glucose saturation effect observed at 10 mM, we believe there might be a misinterpretation. We believe that organisms modulate metabolic reactions underlying carbon-nitrogen balance based on the availability of both substrates and the capacities of associated enzymes. An influx of a single substrate (like glucose in our context) can exhaust the other substrate (in this case, ammonium) and/or the enzymatic capacity (GS, GOGAT, etc.), which likely accounts for the observed saturation effect, rather than any direct signaling mechanisms.

Any further exploration of the potential involvement of Wnt, or any other signaling pathway, would require formulating a testable hypothesis and performing adequate experiments to test it. As mentioned above, this study and experiments were not designed for hypothesis generation and exploration.

To enhance the clarity, we have made the following changes:

“Gene ontology (GO) assisted functional analysis of gene expression changes showed that glucose-induced genes were enriched primarily in amino acid biosynthesis, transcription regulation, and translation processes in both symbiotic and aposymbiotic anemones (Figure 3a, b). The canonical Wnt signaling pathway, well-known for its function in cell proliferation²⁴, was activated in symbiotic *E. diaphana* supplied with glucose. This might indicate an overall upregulation of biological processes and pathways involved in cell growth upon glucose provision. Subsequent pathway enrichment analyses showed that glucose supplementation specifically induced the expression of genes associated with nitrogen metabolism and amino acid biosynthesis in both symbiotic and aposymbiotic *E. diaphana* (Figure 3c). Further integration of these enriched biological processes and pathways highlighted the GS/GOGAT-mediated amino acid biosynthesis pathway that we identified previously¹¹ and all of the genes associated with this pathway were upregulated in symbiotic anemones compared to aposymbiotic ones when no glucose was supplied (Figure 3d).”

Line 205: not just Aiptasia

Figure 4: were the scales log₂(fold change) in the heatmaps? This needs to be clarified.

We are grateful to the reviewer for pointing out these areas of improvement.

For line 205, we acknowledge the oversight and have made the necessary edits to rectify it.

“Figure 4 Identification of isotope-labeled metabolites using UHPLC-HR-MS.”

We have also made changes in line 194, where the figure is referenced, to enhance precision and clarity.

“Here we present the identification of glutamine from *E. diaphana* as an example (Figure 4a).”

Regarding Figure 4, the reviewer's query is well-noted. The heatmaps portray the relative abundance of isotopologues containing isotopes, specifically summarized as ¹³C¹⁴N, ¹²C¹⁵N, and ¹³C¹⁵N, which are then normalized against their non-labeled counterparts (¹²C¹⁴N). We have now revised the figure caption to ensure clarity for readers.

“Heatmap color indicates the relative abundance of isotope-labeled metabolites (specifically summarized as $^{13}\text{C}^{14}\text{N}$, $^{12}\text{C}^{15}\text{N}$, and $^{13}\text{C}^{15}\text{N}$) normalized to their non-labeled counterparts ($^{12}\text{C}^{14}\text{N}$).”

Line 231-233: “ In particular, most of the ^{15}N isotope was identified in both ^{13}C - and ^{15}N -containing metabolites ($^{13}\text{C}^{15}\text{N}$) from animals with the combined treatment, while only a small proportion of the ^{15}N isotope ended up in $^{12}\text{C}^{15}\text{N}$ compounds ”. This data appeared to confirm that the supplemented glucose and ammonium were metabolized into ^{13}C - and ^{15}N -containing metabolites. In contrast, only a small portion of ^{12}C , which was potentially from algal photosynthesis, appeared to be incorporated into $^{12}\text{C}^{15}\text{N}$ compounds... There was some accumulation of $^{12}\text{C}^{15}\text{N}$ metabolites under the condition supplemented with ^{15}N , but similar in Aiptasia-Apo... So it was unclear about the incorporation of the photosynthetically derived carbon. In any case, the authors may need to clarify the data interpretation from this experimental design..

We appreciate the reviewer’s comments on this section.

The observed $^{12}\text{C}^{15}\text{N}$ in aposymbiotic Aiptasia is likely a reflection of the anemone’s intrinsic nitrogen metabolism. In ^{15}N -ammonium incubation, the baseline metabolism incorporates ^{15}N , leading to its appearance in the amino acids, as shown in our experiments. It’s crucial to note that this presence does not necessarily represent a net increase in total N, as we lack nitrogen efflux data to determine a comprehensive uptake.

It is clear that the addition of carbon distinctly enhances nitrogen assimilation. This amplified effect is evident in both symbiotic Exaiptasia, where carbon is sourced from the symbiont’s photosynthesis, and in scenarios with glucose supplementation.

With ^{15}N -ammonium treatment alone, there’s a pronounced $^{12}\text{C}^{15}\text{N}$ abundance in symbiotic anemones in comparison to aposymbiotic ones. This likely reflects a boost in host ammonium assimilation resulting from the additional symbiont’s photosynthates.

In the combined treatment of ^{15}N -ammonium and ^{13}C -glucose, while there’s a notable increase in $^{13}\text{C}^{15}\text{N}$ due to the added glucose, a proportion of the ^{15}N still integrates into $^{12}\text{C}^{15}\text{N}$. Notably, this $^{12}\text{C}^{15}\text{N}$ fraction is more abundant in symbiotic anemones than in aposymbiotic ones, further emphasizing the symbionts’ contribution to the host’s ammonium assimilation.

“In addition, significant increases in ^{15}N incorporation rates were also observed when additional carbon was present. In the case where carbon is sourced from symbiont photosynthesis, symbiotic *E. diaphana* assimilated more ^{15}N compared to aposymbiotic anemones, highlighting the role of symbiont-derived photosynthates in host ammonium assimilation. In scenarios of glucose supplementation, ^{15}N incorporation rates were further enhanced. In particular, most of the ^{15}N isotope was identified in both ^{13}C - and ^{15}N -containing metabolites ($^{13}\text{C}^{15}\text{N}$) from animals with the combined treatment, while only a small proportion of the ^{15}N isotope ended up in $^{12}\text{C}^{15}\text{N}$ compounds (Tables S1-S4), which indicates that most of the ^{15}N was assimilated through the incorporation into carbon backbones derived from the $^{13}\text{C}_6$ -glucose provided. This finding further supported the hypothesis that the metabolization of glucose to 3-phosphohydroxypyruvate produces the carbon backbones required for ammonium assimilation through the GS/GOGAT cycle.”

Reviewer #2 (Remarks to the Author):

The authors have adequately addressed my concerns with the manuscript.

We thank the reviewer for acknowledging our efforts. We truly appreciate the reviewer’s constructive feedback and are pleased to have addressed all these points.

Reviewer #3 (Remarks to the Author):

I have no further comments on the manuscript. All my points are answered comprehensively and satisfactorily.

We appreciate the reviewer's feedback. And we are glad that our responses have addressed all the reviewer's points. The reviewer's insights were invaluable in improving our manuscript.